# PRIVACY AMPLIFICATION FOR MATRIX MECHANISMS

**Christopher A. Choquette-Choo**[*]    **Arun Ganesh**[†]    **Thomas Steinke**[‡]
**Abhradeep Thakurta**[§]

## ABSTRACT

Privacy amplification exploits randomness in data selection to provide tighter differential privacy (DP) guarantees. This analysis is key to DP-SGD's success in machine learning (ML), but, is not readily applicable to the newer state-of-the-art (SOTA) algorithms. This is because these algorithms, known as DP-FTRL, use the *matrix mechanism to add correlated noise* instead of independent noise as in DP-SGD.

In this paper, we propose "MMCC", the first algorithm to analyze privacy amplification via sampling for any generic matrix mechanism. MMCC is nearly tight in that it approaches a lower bound as $\varepsilon \to 0$. To analyze correlated outputs in MMCC, we prove that they can be analyzed as if they were independent, by conditioning them on prior outputs. Our "conditional composition theorem" has broad utility: we use it to show that the noise added to binary-tree-DP-FTRL can asymptotically match the noise added to DP-SGD with amplification. Our algorithm also has practical empirical utility. We show that amplification leads to significant improvement in the privacy/utility trade-offs for DP-FTRL style algorithms for standard benchmark tasks.

## 1 INTRODUCTION

Privacy amplification is key in differentially private (DP) machine learning (ML) as it enables tighter privacy budgets under certain assumptions on the data processing. For example, one of the main contributions in the DP-SGD (DP Stochastic Gradient Descent) work by Abadi et al. (2016) was the "moments accountant", which relies on privacy amplification (Kasiviswanathan et al., 2008; Bassily et al., 2014) for bounding the privacy cost. Recently, privacy amplification analysis enabled Choquette-Choo et al. (2023a) to show that a class of DP-FTRL (DP Follow-The-Regularized-Leader) algorithms (Smith & Thakurta, 2013; Kairouz et al., 2021) was superior in privacy-utility tradeoffs to DP-SGD.[1] At the heart of DP-FTRL is the construct of matrix mechanism (McKenna et al., 2021; Denisov et al., 2022) that effectively computes the prefix sums $\sum_{i \leq t} \mathbf{x}_i$ over a sequence of adaptively chosen vectors $\{\mathbf{x}_i : i \in [n]\}$ (given by $\mathbf{A} \cdot \mathbf{x}$, where $\mathbf{A}$ is a lower triangular matrix of all ones and $\mathbf{x} = [\mathbf{x}_1|\cdots|\mathbf{x}_n]^\top \in \mathcal{R}^{n \times d}$). Matrix mechanism corresponds to factorizing $\mathbf{A} = \mathbf{B} \cdot \mathbf{C}$ to minimize the error in the estimation of the prefix sums, while ensuring $\mathbf{C} \cdot \mathbf{x} + \mathbf{z}$ satisfies DP, where $\mathbf{z}$ is drawn from an isotropic normal distribution. Bringing privacy amplification to matrix mechanisms is thus an important area of research to enable better privacy-utility tradeoffs. (In the rest of the paper, we will refer to the matrix $\mathbf{B}$ as the decoder matrix, and $\mathbf{C}$ as the encoder matrix.)

[*]Google DeepMind. `cchoquette@google.com`.

[†]Google Research. `arunganesh@google.com`.

[‡]Google DeepMind. `steinke@google.com`.

[§]Google DeepMind. `athakurta@google.com`.

[1]Precisely, they showed DP-FTRL is never worse, and often better, than DP-SGD—it "pareto-dominates".

Matrix mechanism poses a major challenge for privacy amplification analysis of DP-FTRL style algorithms. Standard privacy amplification exploits randomness in the selection of minibatches[2] but requires that the noise added to each minibatch is independent. In the matrix mechanism, a minibatch (given by $\mathbf{x}_i$) contributes to multiple rows of $\mathbf{C} \cdot \mathbf{x} + \mathbf{z}$, thus preventing direct application of amplification. This challenge can be seen by the limitations of the amplification analysis of Choquette-Choo et al. (2023a) which only applies to a special class of '$b$-banded' matrix mechanisms (i.e., the first $b$ principal diagonals of $\mathbf{C}$ are non-zero), that in-turn leads to multiplicatively higher sampling probabilities preventing the full benefits of amplification. Resulting from these limitations is a large region of epsilons where banded matrix mechanisms cannot simultaneously leverage the benefits of correlated noise and privacy amplification; in other words, they perform equivalent to, *but no better than, DP-SGD*[3]. Further, since their analysis only applies to the special banded case, matrix mechanisms from the extant literature cannot leverage amplification and correlated noise, e.g., Kairouz et al. (2021); Choquette-Choo et al. (2023b); Denisov et al. (2022).

*In this work, we provide a generic privacy amplification machinery for adaptive matrix mechanisms for any lower-triangular encoder matrix $\mathbf{C}$ that strictly generalizes the approach in (Choquette-Choo et al., 2023a).*

## 1.1 OUR CONTRIBUTIONS

Our main contribution is to prove a general privacy amplification analysis for *any matrix mechanism*, i.e., arbitrary encoder matrices $\mathbf{C}$ for *non-adaptively* chosen $\mathbf{x}$, and for lower-triangular $\mathbf{C}$'s when $\mathbf{x}$ is *adaptively* chosen (which is the typical situation for learning tasks). We then demonstrate that our method yields both asymptotic improvements and experimental improvements.

**Conditional composition (Sec. 3, Theorem 3.1):** This is our main technical tool that gracefully handles dependence between the queries $\mathbf{C}_{[i,:]} \cdot \mathbf{x}$, and $\mathbf{C}_{[i+1,:]} \cdot \mathbf{x}$ that arises due to multiple participation of a single row of the data matrix $\mathbf{x}$. Standard composition theorems (Dwork & Roth, 2014) only handle this via a *pessimistic worst-case privacy guarantee* that holds with certainty for each query (in our application, a query is $\mathbf{C}_{[i,:]} \cdot \mathbf{x} + \mathbf{z}_i$ conditioned on $\mathbf{C}_{[1:i-1,:]} \cdot \mathbf{x} + \mathbf{z}_{1:i-1}$). Theorem 3.1 relaxes this to holding with high probability (over the randomness of the algorithm) leading to significantly better guarantees. This generalizes an idea previously used in (Erlingsson et al., 2019; Balle et al., 2020) to analyze privacy amplification by shuffling. We believe this theorem will be useful for analyzing correlated noise mechanisms beyond those studied herein.

**Matrix mechanism with uniform sampling via `MMCC` (Sec. 4):** We prove amplified privacy guarantees for the matrix mechanism with uniform sampling, using Theorem 3.1, that are nearly-tight in the low-epsilon regime (as $\varepsilon \to 0$). We improve over Choquette-Choo et al. (2023a) because we enable "more randomness" in sampling—instead of participating w.p. $bp$ in $n/b$ rounds, data records can participate w.p. $p$ in all $n$ rounds.

Recall we need to analyze the privacy of outputting $\mathbf{C}\mathbf{x} + \mathbf{z}$, where rows of $\mathbf{x}$ are chosen via uniform sampling. We use Thm. 4.3 to reduce $\mathbf{C}\mathbf{x} + \mathbf{z}$ to a series of *mixture of Gaussians (MoG) mechanisms* for which we can use privacy loss distribution (PLD) accounting (see the proof sketch of Thm. 4.3 in Sec. 4). Our algorithm `MMCC` formally stating this reduction is given in Fig. 1. We plan to publicly release a library implementing `MMCC` with the final manuscript. The analysis of MoG mechanisms included in this library has other uses, such as tighter privacy guarantees for DP-SGD with group-level DP or for linear losses, see App. F for more discussion.

---

[2]E.g., for a data set $D$, a row $\mathbf{x}_{[i,:]} = \sum_{d \in S} \nabla_\theta \ell(\theta_i; d)$, where $S$ is a randomly chosen subset of $D$ (e.g., sampled u.a.r. from $D$, or a subset from a random shuffling of $D$), $\ell$ is a loss function, and $\theta_i$ is obtained via an SGD state update process.

[3]In Choquette-Choo et al. (2023a), this region surfaces empirically even for larger $\varepsilon \approx 1$.

**Binary tree analysis (Sec. 5):** Letting $\sigma_{\varepsilon,\delta}$ be the noise required for the Gaussian mechanism to achieve to satisfy $(\varepsilon, \delta)$-DP, the binary tree mechanism requires noise $\sigma_{\varepsilon,\delta} \cdot \sqrt{\log n}$. Owing to the versatility of conditional composition, we show that with shuffling, the (non-adaptive) binary tree mechanism only needs noise $\sigma_{\varepsilon,\delta} \cdot \min\{\sqrt{\log n}, \sqrt{\log\log(1/\delta)}\}$. This is optimal given current amplification by shuffling results, which require $n = \Omega(\log 1/\delta)^4$. To the best of our knowledge, this is the first amplification guarantee (of any kind) for the binary tree mechanism.

**Empirical improvements (Sec. 6):** First we implement and show that $\varepsilon$ computed via MMCC for the binary tree mechanism matches the theoretical predictions of $\Omega(\sqrt{\log n})$ from Sec. 5. Then we apply our work to machine learning and show we can improve the privacy-utility tradeoff for binary-tree-DP-FTRL (Kairouz et al., 2021) entirely post-hoc. Finally, we show that the "every round" sampling enabled by MMCC achieves better amplification than the "$b$-min-sep" sampling of (Choquette-Choo et al., 2023a).

## 1.2 PROBLEM DEFINITION

**Matrix mechanism (MM):** Consider a workload matrix $\mathbf{A} \in \mathcal{R}^{n \times n}$, and consider a data set $D = \{d_1, \ldots, d_m\} \in \mathcal{D}^m$. Let $\mathbf{x} = [\mathbf{x}_1(D) | \cdots | \mathbf{x}_n(D)]^\top \in \mathcal{R}^{n \times d}$ be a matrix s.t. each row $\mathbf{x}_i : \mathcal{D}^* \to \mathcal{R}^d$ is a *randomized* function that first selects a subset of the data set $D$, and then maps it to a real valued vector. Furthermore, each of the $\mathbf{x}_i$ has the following two properties. a) *Decomposability*: For the subset of the data set $D$ that $\mathbf{x}_i$ chooses (call it $S_i$), we have $\mathbf{x}_i(D) = \sum_{d \in S_i} \mathbf{g}_i(d)$ with $\mathbf{g}_i : \mathcal{D} \to \mathcal{R}^d$ is a vector valued function, and b) *bounded sensitivity*: $\forall d \in \mathcal{D} : \|\mathbf{g}_i(d)\|_2 \leq 1$. Matrix mechanism corresponds to the class of DP algorithms that approximates $\mathbf{A}\mathbf{x}$ with low-error. Typically, one designs a pair of matrices $\mathbf{A} = \mathbf{B}\mathbf{C}$ ( which we will call the *decoder* and the *encoder* matrices respectively) s.t. $\mathbf{C}\mathbf{x} + \mathbf{z}$ satisfies DP[5] (with $\mathbf{z}$ being isotropic normal noise), and $\mathbf{B}\mathbf{z}$ is minimized in appropriately chosen norm. We will assume $\mathbf{C}$ is non-negative for simplicity.

**Privacy amplification for MM:** In this work we study the problem of amplifying the DP guarantee of MM if we incorporate the randomness in the selection of the subsets of $D$ by each function $\mathbf{x}_i$. In particular we consider two selection strategies: i) *uniform sampling*: each $\mathbf{x}_i$ selects each entry of $D$ independently w.p. $p$, ii) *shuffling*: First the records of $D$ are randomly permuted, and then each $\mathbf{x}_i$ picks a fixed disjoint subset (of equal size) from $D$.

**Adaptivity:** In our work we allow the choice of $\mathbf{x}_i$'s to be adaptive, i.e., $\mathbf{x}_i$ can be chosen based on the first $i-1$ outputs of MM. Under adaptivity, we will only consider encoder ($\mathbf{B}$) decoder matrices ($\mathbf{C}$) that are lower triangular. However, for non-adaptive choices of the $\mathbf{x}_i$'s we allow arbitrary choice of the matrices $\mathbf{B}$ and $\mathbf{C}$. *Unless mentioned specifically, all our results will be for the adaptive setting.*

## 2 BACKGROUND AND RELATED WORKS

### 2.1 PRIVACY LOSS DISTRIBUTIONS (PLD)

Suppose we have a DP mechanism $\mathcal{M}$ that outputs a sample from the continuous distribution $P = \mathcal{M}(D)$ when given database $D$, and outputs a sample from $Q = \mathcal{M}(D')$ when given $D'$. The $\varepsilon$-hockey stick divergence between two distributions $P, Q$ is defined as:

$$H_\varepsilon(P, Q) = \int_x \max\{P(x) - e^\varepsilon Q(x), 0\} \mathrm{d}x = \mathbb{E}_{x \sim P}\left[\max\left\{1 - \frac{e^\varepsilon}{e^{\ln(P(x)/Q(x))}}, 0\right\}\right].$$

---

[4]We believe this requirement is fundamental and thus $\sigma_{\varepsilon,\delta} \cdot \min\{\sqrt{\log n}, \sqrt{\log\log(1/\delta)}\}$ is optimal, but if it were removed, our result would improve to $O(1) \cdot \sigma_{\varepsilon,\delta}$.

[5]We use the zero-out adjacency (Ponomareva et al., 2023) to define DP in this paper.

A mechanism $\mathcal{M}$ satisfies $(\varepsilon, \delta)$-DP if and only if for all adjacent databases $D, D'$ we have $H_\varepsilon(\mathcal{M}(D), \mathcal{M}(D')) \leq \delta$. From the definition, we see that to obtain the $\varepsilon$-hockey stick divergence between $P$ and $Q$, it suffices to know their *privacy loss distribution* (PLD):

**Definition 2.1.** *The privacy loss random variable for $P$ and $Q$ is given by sampling $x \sim P$, and computing* $\ln(P(x)/Q(x))$. *The PLD of $P$ and $Q$ is the distribution of this random variable.*

We frequently use the notion of dominating PLDs:

**Definition 2.2** (Definition 7 in (Zhu et al., 2022)). *The PLD of $P, Q$ dominates the PLD of $P', Q'$ if for any* $\varepsilon, H_\varepsilon(P, Q) \geq H_\varepsilon(P', Q')$. *We will also say random variable $L$ dominates random variable $L'$ if for any* $\varepsilon, H_\varepsilon(L) \geq H_\varepsilon(L')$, *where $H_\varepsilon(L) = \mathbb{E}_{\ell \sim L} \left[\max\left\{1 - e^{\varepsilon - \ell}, 0\right\}\right]$.*

Informally, a PLD dominates another PLD if any privacy guarantee satisfied by mechanisms with the dominating PLD is also satisfied by mechanisms with the dominated PLD. In particular, if the PLD of some pair of distributions $P, Q$ dominates the PLDs of all pairs $\mathcal{M}(D), \mathcal{M}(D')$ for adjacent $D, D'$, then if $H_\varepsilon(P, Q) \leq \delta$, $\mathcal{M}$ satisfies $(\varepsilon, \delta)$-DP.

## 2.2 PRIVACY AMPLIFICATION

Privacy amplification via sampling analyzes the improvement in privacy given by randomly sampling a minibatch of examples instead of choosing it deterministically. Roughly, a $(\varepsilon, \delta)$-DP mechanism run on a batch where each example participates with probability $p$ satisfies $(\log(1 - p + pe^\varepsilon), \delta)$-DP. The relative improvement from $\varepsilon$ to $\log(1 - p + pe^\varepsilon)$ gets better as $\varepsilon$ gets smaller: $\log(1 - p + pe^\varepsilon) \approx p\varepsilon$ for $\varepsilon < 1$, but $\log(1 - p + pe^\varepsilon) \approx \varepsilon - \log(1/p)$ for large $\varepsilon$. The benefits of privacy amplification via sampling in the independent noise setting of DP-SGD, i.e., the decoder matrix $\mathbf{C} = \mathbb{I}$, are extremely well-studied (Song et al., 2013; Bassily et al., 2014; Abadi et al., 2016; Mironov et al., 2019; Steinke, 2022; Koskela et al., 2020) with tight analyses. In particular, one round of DP-SGD is dominated by the PLD of $N(0, \sigma^2)$ and $(1-p) \cdot N(0, \sigma^2) + p \cdot N(1, \sigma^2)$ and since each round of DP-SGD has independent randomness, composing this PLD with itself $n$ times gives a tight dominating PLD, i.e. tight $(\varepsilon, \delta)$ curve, for DP-SGD.

## 3 CONDITIONAL COMPOSITION

We first show a conditional composition theorem, which allows us to analyze a sequence of adaptive mechanisms using high-probability instead of worst-case privacy guarantees for each mechanism. We state conditional composition formally as Theorem C.2. This is a generalization of an idea used in (Erlingsson et al., 2019; Balle et al., 2020) to analyze amplification by shuffling.

**Theorem 3.1** (Informal version of Theorem C.2). *Let $\mathcal{M}_1, \mathcal{M}_2, \ldots$ be a sequence of adaptive mechanisms, where each $\mathcal{M}_i$ takes $D$ and the previous mechanisms' output as input. Suppose there is some "bad" event $E$ that happens with probability at most $\delta_{bad}$ over the randomness of $\mathcal{M}_1, \mathcal{M}_2, \ldots$ for any input $D$. If the composition of the worst-case privacy guarantees of $\mathcal{M}_1, \mathcal{M}_2, \ldots$ each conditioned on $E$ not happening satisfies $(\varepsilon, \delta)$-DP, then the composition of $\mathcal{M}_1, \mathcal{M}_2, \ldots$ satisfies $(\varepsilon, \delta + \delta_{bad})$-DP.*

The proof is given in App. C. To apply Theorem 3.1 to correlated noise mechanisms, we observe that they can be viewed as a sequence of adaptive independent-noise mechanisms:

**Observation 3.2.** *Let $\mathcal{M} : \mathcal{D} \to \mathcal{X}_1 \times \mathcal{X}_2 \times \ldots \times \mathcal{X}_n$ be a mechanism that takes a dataset $D$ and outputs the tuple $x = (x_1, x_2, \ldots, x_n)$ drawn from the distribution $\mathcal{M}(D)$. Let $\mathcal{M}_i : \mathcal{X}_1 \times \mathcal{X}_2 \times \ldots \times \mathcal{X}_{i-1} \times \mathcal{D} \to \mathcal{X}_i$ be the mechanism that takes $x'_1, x'_2, \ldots, x'_{i-1}$ and a dataset $D$ and outputs $x'_i$ with probability (or likelihood) $\mathbf{Pr}_{x \sim \mathcal{M}(D)} \left[x_i = x'_i | x_1 = x'_1, x_2 = x'_2, \ldots, x_{i-1} = x'_{i-1}\right]$. The output distributions of $\mathcal{M}$ and the composition of $\mathcal{M}_1, \mathcal{M}_2, \ldots$ are the same.*

---

**Algorithm 1** **M**atrix **M**echanism **C**onditional **C**omposition algorithm, $\mathrm{MMCC}(\mathbf{C}, p, \sigma, \delta_1, \delta_2)$

---

1: **Input:** Matrix $\mathbf{C}$, sampling probability $p$, noise standard deviation $\sigma$, probabilities $\delta_1, \delta_2$.
2: $\{\widetilde{p}_{i,j}\}_{i,j \in [n]} \leftarrow \mathtt{ProbabilityTailBounds}(\mathbf{C}, p, \sigma, \delta_1)$.
   $\triangleright$ $\widetilde{p}_{i,j}$ is a high-probability upper bound on the probability that an example participated in round $j$, conditioned on output in rounds 1 to $i-1$.
3: **for** $i \in [n]$ **do**
4:     $PLD_i \leftarrow$ PLD of $\mathcal{M}_{PMoG}(\{\widetilde{p}_{i,j}\}_{j \in [n]}, \{\mathbf{C}_{i,j}\}_{j \in [n]})$.
5: $PLD \leftarrow$ convolution of $\{PLD_i\}_{i \in [n]}$.
6: **return** $\min\left(\{\varepsilon : PLD \text{ satisfies } (\varepsilon, \delta_2)\text{-DP}\}\right)$.

---

Figure 1: Algorithm $\mathrm{MMCC}$ for computing amplified privacy guarantees of the matrix mechanism. The subroutine $\mathtt{ProbabilityTailBounds}$ is given in Fig. 5 in App. C.

## 4 PRIVACY ANALYSIS FOR MATRIX MECHANISMS

In this section, we give an algorithm for computing an upper bound on the privacy guarantees of the matrix mechanism, and prove its correctness.

### 4.1 MIXTURE OF GAUSSIANS MECHANISMS

The key tool in our privacy analysis is a *mixture of Gaussians mechanism*, a generalization of the Gaussian mechanism with sampling. Here we define these mechanisms under the add adjacency, i.e. $D'$ contains an example zeroed out in $D$.

**Definition 4.1.** *A **mixture of Gaussians (MoG) mechanism** is defined by two lists, a list of probabilities $\{p_1, p_2, \ldots, p_k\}$, with $\sum_i p_i = 1, p_i \in [0,1]$, a list of sensitivities $\{c_1, c_2, \ldots c_k\}$ and a noise level $\sigma$. For simplicity, we will assume $c_i \geq 0$. Given $D$, the mechanism $\mathcal{M}_{MoG}(\{p_1, p_2, \ldots, p_k\}, \{c_1, c_2, \ldots c_k\})$ outputs $z \sim N(0, \sigma^2)$. Given $D'$, it samples $s$ from the distribution with support $\{c_i\}_{i \in [k]}$ and associated probabilities $\{p_i\}_{i \in [k]}$, and outputs $z \sim N(s, \sigma^2)$. In other words, it is a Gaussian mechanism where the sensitivity $s$ is a random variable distributed according to $\{p_i\}_{i \in [k]}, \{c_i\}_{i \in [k]}$.*

*A **vector mixture of Gaussians (VMoG) mechanism** $\mathcal{M}_{VMoG}$ is the same as a MoG mechanism, except the sensitivities $c_i$ are allowed to be vectors $\mathbf{c}_i$ instead of scalars, and our output is sampled from a multivariate Gaussian $\mathbf{z} \sim N(\mathbf{0}, \sigma^2 \cdot \mathbb{I})$ or $\mathbf{z} \sim N(\mathbf{s}, \sigma^2 \cdot \mathbb{I})$.*

For our proofs, we will need to prove a few properties of MoG mechanisms. We give these properties and their proofs in App. B. It will be easier for us to work with a special case of MoG mechanisms, where the probabilities and sensitivities arise from a product distribution:

**Definition 4.2.** *A **product mixture of Gaussians (PMoG) mechanism** is defined by two lists $\{p_1, \ldots, p_k\}$ and $\{c_1, \ldots c_k\}$ and a noise level $\sigma$. The mechanism $\mathcal{M}_{PMoG}(\{p_1, \ldots, p_k\}, \{c_1, \ldots, c_k\})$ is defined equivalently as $\mathcal{M}_{MoG}(\{\prod_{i \in S} p_i \cdot \prod_{i \notin S}(1 - p_i) | S \in 2^{[k]}\}, \{\sum_{i \in S} c_i | S \in 2^{[k]}\})$.*

### 4.2 MATRIX MECHANISM CONDITIONAL COMPOSITION

The high-level idea of our algorithm, $\mathrm{MMCC}$ (short for *matrix mechanism conditional composition*), for analyzing the matrix mechanism with amplification is the following: The output of each round conditioned on the previous rounds' output is a MoG mechanism. For each round, we specify a MoG mechanism that dominates this MoG mechanism with high probability. Then by Theorem 3.1, it suffices to compute the

privacy loss distribution of each of the dominating MoGs, and then use composition to get our final privacy guarantee. MMCC is given in Fig. 1. In App. C, we prove that MMCC computes a valid DP guarantee:

**Theorem 4.3.** *Let $\varepsilon$ be the output of MMCC$(\mathbf{C}, p, \sigma, \delta_1, \delta_2)$. The matrix mechanism with matrix $\mathbf{C}$, uniform sampling probability $p$, and noise level $\sigma$ satisfies $(\varepsilon, \delta_1 + \delta_2)$-DP.*

We give a high-level overview of the proof. The proof proceeds in three steps. First, we show the matrix mechanism is dominated by a sequence of adaptively chosen *scalar* MoG mechanisms, by analyzing the distribution for each round conditioned on previous rounds and applying a vector-to-scalar reduction (Lem. B.4). Second, we simplify these MoG mechanisms by showing that each is dominated by a PMoG mechanism with probabilities $p_{i,j}$ depending on the outputs from previous rounds. Third, we show that with high probability $p_{i,j} \leq \widetilde{p}_{i,j}$ for all $i, j$, i.e., the upper bounds generated by ProbabilityTailBounds hold. We then apply Theorem 3.1.

**Tightness:** To get a sense for how tight MMCC is, if in MMCC we instead set $\widetilde{p}_{i,j} = p$ for all $i, j$, this is equivalent to analyzing the matrix mechanism as if each row were independent. Since the rows are actually correlated, we expect this analysis to give a lower bound on the true value of $\varepsilon$. So we can use $\max_{i,j} \widetilde{p}_{i,j}/p$ as roughly an upper bound on the ratio of the $\varepsilon$ reported by MMCC and the true $\varepsilon$ value. In particular, as $\sigma \to \infty$, for $\widetilde{p}_{i,j}$ computed by ProbabilityTailBounds this ratio approaches 1, i.e. MMCC gives tight $\varepsilon$ guarantees in the limit as $\sigma \to \infty$.

**Sampling scheme of (Choquette-Choo et al., 2023a):** The techniques used in MMCC are complementary to those in (Choquette-Choo et al., 2023a): In App. D, we give a generalization of MMCC that analyzes the matrix mechanism under their "$b$-min-sep sampling." For $b = 1$, this is the same as i.i.d. sampling every round so this generalization retrieves MMCC. For $b$-banded matrices this generalization retrieves exactly the DP-SGD-like analysis of (Choquette-Choo et al., 2023a). In other words, *this generalization subsumes all existing amplification results for matrix mechanisms.*

**Benefits of i.i.d. sampling:** MMCC is the first analysis that allows us to benefit from both correlated noise and privacy amplification via i.i.d. (i.e., maximally random) sampling. In Sec. 6.3 we demonstrate that the combination of benefits allows us to get better $\ell_2^2$-error for computing all prefix sums than independent-noise mechanisms, for much smaller $\varepsilon$ than prior work.

## 5 AMPLIFICATION VIA SHUFFLING FOR NON-ADAPTIVE BINARY TREE

In this section, we show that amplification allows us to improve the privacy guarantees of the binary tree mechanism of (Dwork et al., 2010; Chan et al., 2011). We consider the setting where first the data set $D$ is randomly permuted (call it $\Pi(D)$), and each function $\mathbf{x}_i$ (in the definition of MM from Section 1.2) picks the $i$-th data record in $\Pi(D)$. Roughly speaking, using privacy amplification by shuffling (see Section 1.2) we improve $\sigma$ for this mechanism by $\Omega(\sqrt{\log n}/\sqrt{\log \log(1/\delta)})$, while maintaining that each example participates once. For simplicity throughout the section we restrict to the case where $n$ is a power of 2.

**Binary tree mechanism:** The binary tree computes sums of rows of $\mathbf{x}$ over the intervals $[1 : 1], [2 : 2], \ldots, [n : n], [1 : 2], [3 : 4], \ldots, [n - 1 : n], [1 : 4], \ldots [1 : n]$ with noise. That is, it outputs $\left\{ \sum_{k \cdot 2^j + 1 \leq i \leq (k+1) \cdot 2^j} \mathbf{x}_i + \mathbf{z}_{j,k} \right\}_{0 \leq j \leq \log n, 0 \leq k < n/2^j}$, where $\mathbf{z}_{j,k} \stackrel{i.i.d.}{\sim} N(0, \sigma^2)$. Equivalently, it is a (non-square) matrix mechanism where for each $j, k$ pair there is a row of $\mathbf{C}$ where the entries in the interval $[k \cdot 2^j + 1 : (k+1) \cdot 2^j]$ are 1 and the remaining entries are 0. We refer to all the noisy sums indexed by the same $j$ as level $j$. In the single-epoch setting (without shuffling), each row of $\mathbf{x}_i$ is a sensitivity-1 function computed on the $i$th example in $D$. The binary tree mechanism then satisfies the privacy guarantees of distinguishing $\mathbf{z}$ and $\mathbf{Ce}_i + \mathbf{z}$, where $\mathbf{e}_i$ is an elementary vector. Since each row of $\mathbf{x}$ is included in $\log n + 1$ of the

sums, we have $\|\mathbf{Ce}_i\|_2 = \sqrt{\log n + 1}$, i.e. the binary tree mechanism satisfies $\left(O\left(\frac{\sqrt{\log(n)\log(1/\delta)}}{\sigma}\right), \delta\right)$-DP.

We now analyze the binary tree mechanism under shuffling. To apply Theorem 3.1, we need the following analysis of "approximate shuffling" given in Lem. 5.1, proven in App. C.

**Lemma 5.1** (Simplification of Lem. C.4 in App. C). *Suppose we run $n$ Gaussian mechanisms on $n$ inputs, where the order of the inputs is chosen according to a distribution such that no input appears in a certain position with probability more than $1/n'$. Then for $\delta \geq 2^{-\Omega(n')}, \delta_0 \geq 0$, this set of mechanisms satisfies* $\left(O\left(\frac{\sqrt{\ln(1/\delta_0)\ln(1/\delta)}}{\sigma\sqrt{n'}}\right), \delta + n'\delta_0\right)$-*DP.*

**Theorem 5.2.** *The non-adaptive binary tree mechanism run on $\Pi(D)$ satisfies* $\left(O\left(\frac{\sqrt{\log(1/\delta)\log\log(1/\delta)}}{\sigma}\right), \delta\right)$-*DP for $\sigma = \Omega(\sqrt{\log(1/\delta)\log\log(1/\delta)})$, $\delta \in [2^{-\Omega(n)}, 1/n]$.*

The proof of Theorem 5.2 is given in App. C. We give a summary here: For the "top" levels $j \in [\log n - O(\log\log(1/\delta)), \log n]$, the number of sums per level (i.e., $n'$ in Lem. 5.1) is less than $\log(1/\delta)$ so we cannot apply Lem. 5.1. Instead we use the unamplified privacy analysis of the Gaussian mechanism to analyze the privacy of the corresponding noisy sums. For the remaining levels, we combine Theorem 3.1 and Lem. 5.1 to show level $j$'s privacy parameter $\varepsilon$ is exponentially decaying in $\log n - j$, i.e. the top $O(\log\log(1/\delta))$ levels dominate the privacy guarantee. Note that the theorem is proven in the non-adaptive case; our argument for adaptivity in Sec. 4 implicitly requires independence of participations across examples, which does not hold for shuffling.

# 6 EMPIRICAL IMPROVEMENTS

We implement MMCC by building on methods in the open-source dp_accounting Python library (DP Team, 2022), and perform empirical studies of the amplification benefits from MMCC. PLD accounting for MoG mechanisms is currently open-sourced as part of the dp_accounting library. We plan to open-source our implementation of MMCC which builds on dp_accounting. There are some challenges in the implementation which we discuss in App. F. For simplicity we use $\delta_1, \delta_2 = \delta/2$ in MMCC.

## 6.1 BINARY TREE MECHANISM AMPLIFICATION

In this section, we show how the privacy guarantee of the binary tree mechanism empirically improves if we use sampling and MMCC. In App. E we repeat this study for a different matrix mechanism proposed by (Fichtenberger et al., 2023).

As a baseline, we fix a constant $c$, and consider the binary tree mechanism under a single-participation constraint, with $\sigma = c\sqrt{\log(n) + 1}$. By the analysis of the Gaussian mechanism, for all $n$ that are powers of 2, the binary tree mechanism with this choice of $\sigma$ under a single-participation constraint without amplification satisfies $(\varepsilon, \delta)$-DP for the same $\varepsilon, \delta$. In other words, as we increase $n$, the privacy guarantee of the unamplified mechanism remains fixed. Then, for the same $c$ and $n$ that are powers of 2, we use MMCC to compute a privacy guarantee for the binary tree mechanism with subsampling probability $1/n$ and the same choice of $\sigma$. By the analyses in Section 5, we expect that with subsampling, the value of $\varepsilon$ will decrease as $\Omega(\sqrt{\log n})$.

In Fig. 2, we observe that empirical improvement in $\varepsilon$ due to amplification is roughly proportional to $\sqrt{\log(n) + 1}$. We also observe two improvements as $c$ (i.e., $\sigma$) increases. First, the multiplicative improvement in $\varepsilon$ increases; second, empirical improvements better match a linear fit to $\sqrt{\log(n) + 1}$. Both

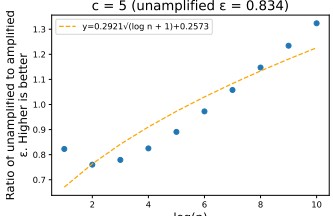 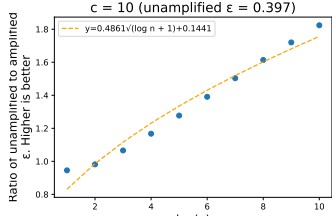 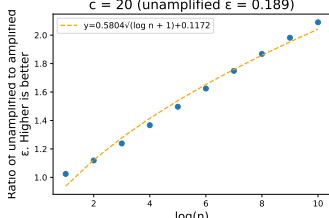

Figure 2: **Multiplicative improvement of our amplification analysis (roughly) matches** $\sqrt{\log(n)+1}$. A higher ratio ($> 1$) indicates amplification is better. We plot $n = 2^i, i \in \{1, 2, \ldots, 10\}$ with $\sigma = c\sqrt{\log(n)+1}$ so $\varepsilon$ is fixed for unamplified single-participation. $\delta = 10^{-6}$.

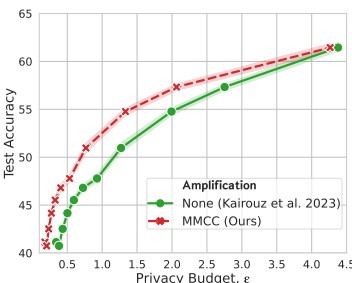

Figure 3: Our amplification analysis leads to significant gains over Kairouz et al. (2021) on practical ML experiments (CIFAR-10), entirely post-hoc.

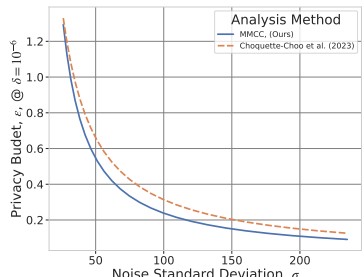

Figure 4: MMCC gives tighter $\varepsilon$ than the analysis of (Choquette-Choo et al., 2023a) for a DP-FTRL-TreeRestart mechanism of height 4. Ran for $n = 512$ steps with $p = \frac{1}{16}$.

these improvements are explained by the fact that (as discussed in Sec. 4) as $\sigma \to \infty$, MMCC reports a tighter $\varepsilon$.

## 6.2 LEARNING EXPERIMENTS WITH BINARY-TREE-DP-FTRL

A motivating reason for us to study matrix mechanisms is that the analysis of Kairouz et al. (2021) has a suboptimal scaling in the amount of noise added, which manifests in their experiments with DP machine learning. We reproduce the centralized DP training on CIFAR-10 from Choquette-Choo et al. (2023b), including model architecture, tuning setup, hyperparameter choices, and optimizations to the tree aggregation mechanism for ML; we use these as our baseline results.

In Fig. 3, we re-analyze the baseline using MMCC and show significant improvements in privacy-utility tradeoffs for DP-FTRL via binary trees. In particular, we observe that these benefits become larger as $\varepsilon$ becomes small. Note that these improvements are entirely "post-hoc," i.e. the algorithm is the same, but with a better privacy analysis.

## 6.3 I.I.D. SAMPLING ENABLES BETTER AMPLIFICATION THAN $b$-MIN-SEP SAMPLING

The prior work of (Choquette-Choo et al., 2023a) gives an amplification result using a sampling scheme we call "$b$-min-sep sampling" for $b$-banded matrices. In their sampling scheme, each example participates in

$n/b$ rounds with sampling probability $bp$. In contrast, MMCC enables sampling each example in all $n$ rounds with probability $p$, a "more random" form of sampling. We compare the two amplification analyses using the DP-FTRL-TreeRestart algorithm of (Kairouz et al., 2021), which sequentially runs $n/2^{h-1}$ height-$h$ binary tree mechanisms, each binary tree mechanism run for $2^{h-1}$ rounds. This corresponds to a matrix mechanism that is $2^{h-1}$-banded, so we can apply the results of (Choquette-Choo et al., 2023a). In Fig. 4, we compare the $\varepsilon$ for DP-FTRL-TreeRestart computed as a function of $\sigma$ using MMCC and the analysis of (Choquette-Choo et al., 2023a), in the setting of $n = 512, p = 1/16, h = 4$, and we see that indeed the more random sampling enabled by MMCC allows for improved privacy guarantees compared to $b$-min-sep sampling.

## 7 DISCUSSION, FUTURE DIRECTIONS, AND CONCLUSION

In this paper, we proposed MMCC, which gives tight amplification guarantees for sampling in the limit as $\varepsilon \to 0$. One limitation of our work is that we are not able to prove adaptivity for non-lower triangular **C**, which captures important matrix mechanisms like the "fully efficient" binary tree mechanism (Honaker, 2015). It is an important future direction to fully understand what combinations of privacy amplification and correlated noise allow the same privacy for non-adaptive and adaptive inputs. In addition, there are many potential improvements to MMCC, as well as open problems that naturally follow from our work. Another open problem that we make progress towards is proving DP-FTRL strictly dominates DP-SGD, i.e. for any $\varepsilon > 0$ DP-FTRL achieves *strictly*[6] better utility than DP-SGD under an appropriate definition of utility. In particular, we conjecture that a tighter amplification analysis than that of MMCC could show that even for $\varepsilon$ close to 0, DP-FTRL with a matrix mechanism where **C** is close to but not equal to the identity has strictly better utility than DP-SGD.

Our interest in the matrix mechanism is primarily motivated by the works of (Denisov et al., 2022; Choquette-Choo et al., 2023b;a) which considered the problem of choosing **C** that optimizes (a proxy for) the utility of DP-FTRL. The utility of DP-FTRL can be written as a function of $\mathbf{C}^{-1}$, and thus can be optimized under a constraint of the form "the matrix mechanism defined by **C** satisfies a given privacy definition". Without amplification, this constraint can usually be easily written as e.g. $\mathbf{C} \in \mathcal{S}$ where $\mathcal{S}$ is a convex set of matrices, which makes optimizing under this constraint easy. An interesting question is whether we can solve the same problem, except the privacy constraint accounts for amplification. This would likely require designing a function that takes $\mathbf{C}, p, \sigma$ and approximates $\varepsilon$ that is differentiable in **C** (unlike MMCC, which is an algorithmic computation that is not easily differentiable).

In these works, DP-FTRL is always strictly better than DP-SGD without amplification, but with amplification for small $\varepsilon$ the optimal choice of **C** with amplification is the identity, i.e. the optimal DP-FTRL is just DP-SGD (with independent noise). If we could optimize **C** under an amplified privacy constraint, we conjecture the following (perhaps surprising) statement could be proven as a corollary: As long as we are not in the full-batch setting, even with amplification by sampling, the optimal choice of **C** is never the identity for $\varepsilon > 0$. In other words, despite its ubiquity, DP-SGD is never the optimal algorithm to use (ignoring computational concerns).

Due to space constraints, we defer the discussion of other future directions to App. A.

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

## A  MORE FUTURE DIRECTIONS

First, our tail bound on the conditional sampling probabilities $\widetilde{p}_{i,j}$ approach $p$ as $\sigma \to \infty$. However, for finite $\sigma$, $\widetilde{p}_{i,j}$ can be much larger than $p$, i.e. the $\varepsilon$ computed by MMCC can be much larger than the true $\varepsilon$. We believe the values of $\widetilde{p}_{i,j}$ we compute are not tight and can be improved. In particular, in computing $\widetilde{p}_{i,j}$, we give a tail bound on the maximum of the dot product of a Gaussian with a set of vectors, and the values of $\widetilde{p}_{i,j}$ we compute effectively correspond to the case where this tail bound is attained by every dot product. This is overly pessimistic, and it should be possible to obtain tighter $\varepsilon$ via a more refined tail-bounding approach.

Second, while MMCC has a polynomial dependence on $n$ (whereas computing $H_\varepsilon$ via e.g. numerical integration would require time exponential in $n$), empirically we found that even with many optimizations for runtime, running MMCC for $n \approx 2000$ still took several hours. In practice, we would often like to run for larger $n$, or do multiple sequential runs of MMCC in order to e.g. compute the smallest $\sigma$ that gives a certain $\varepsilon$ via binary search. In turn, it is practically interesting/important to make MMCC more efficient, or discover another algorithm that gives $\varepsilon$ comparable to or better than MMCC, but with a smaller runtime.

## B  PROPERTIES OF MoG MECHANISMS

The following lemma informally lets us show that domination for the remove adjacency (i.e., $D$ contains an example zeroed out in $D'$) is equivalent to domination for the add adjacency (i.e., $D'$ contains an example zeroed out in $D$). Thus, we usually only need to prove statements under one of the two adjacencies, and it is implied for the other as well.

**Lemma B.1** (Lemma 29 in (Zhu et al., 2022))**.** *The PLD of $P, Q$ dominates the PLD of $P, Q'$ if and only if the PLD of $Q, P$ dominates the PLD of $Q', P$.*

### B.1  MONOTONICITY OF MoG MECHANISMS

The following shows the privacy guarantees of a MoG mechanism are "monotonic" in the sensitivity random variable $\mathbf{c}_i$:

**Lemma B.2.** *Let $\{p_1, p_2, \ldots p_k\}, \{\mathbf{c}_1, \mathbf{c}_2, \ldots, \mathbf{c}_k\}$ and $\{\mathbf{c}_1', \mathbf{c}_2', \ldots \mathbf{c}_k'\}$ be such that (i) each $\mathbf{c}_i$ is non-negative and (ii) $\mathbf{c}_i'$ is entry-wise greater than or equal to $\mathbf{c}_i$ for all $i$, i.e. each $\mathbf{c}_i' - \mathbf{c}_i$ is non-negative.*

*Then the PLD of*

$$\mathcal{M}_{VMoG}(\{p_1, p_2, \ldots p_k\}, \{\mathbf{c}_1, \mathbf{c}_2, \ldots, \mathbf{c}_k\})$$

*is dominated by the PLD of*

$$\mathcal{M}_{VMoG}(\{p_1, p_2, \ldots p_k\}, \{\mathbf{c}_1', \mathbf{c}_2', \ldots \mathbf{c}_k'\}).$$

*Proof.* By Lem. B.1, it suffices to only consider the remove adjacency, i.e. given $D$ we sample $\mathbf{c}_i$ and then sample from $N(\mathbf{c}_i, \sigma^2 \mathbb{I})$ and given $D'$ from $N(\mathbf{0}, \sigma^2 \mathbb{I})$. The privacy loss of outputting $\mathbf{x}$ is:

$$PL(\mathbf{x}) := \ln \left( \sum_i p_i \exp \left( \frac{2\langle \mathbf{c}_i, \mathbf{x} \rangle - \|\mathbf{c}_i\|_2^2}{2\sigma^2} \right) \right).$$

Let $S \subseteq \mathbb{R}^d$ be *monotonic* if for any $\mathbf{x} \in S, \mathbf{y}$ such that $\mathbf{y} - \mathbf{x}$ is non-negative, $\mathbf{y}$ is also in $S$. In other words, increasing any subset of the entries of $\mathbf{x} \in S$ gives another vector in $S$. Since all $\mathbf{c}_i$ are non-negative, if $\mathbf{y} - \mathbf{x}$ is non-negative, then the privacy loss of outputting $\mathbf{y}$ is larger than that of outputting $\mathbf{x}$. So for any

VMoG mechanism and any $t$, the set of outputs $S_t = \{\mathbf{x} : PL(\mathbf{x}) \geq t\}$ is monotonic. By the Neyman-Pearson lemma it suffices to consider only the sets $S_t$ in the definition of $(\varepsilon, \delta)$-DP, i.e. a mechanism satisfies $(\varepsilon, \delta)$-DP if and only if

$$\forall t : \Pr_{x \sim \mathcal{M}(D)}[x \in S_t] \leq e^\varepsilon \cdot \Pr_{x \sim \mathcal{M}(D')}[x \in S_t] + \delta.$$

So, in order to show that to show the first VMoG mechanism is dominated by the second, it suffices to show the probability that $\mathbf{x} \sim N(\mathbf{c}_i, \sigma^2 \mathbb{I})$ is in any monotonic set $S$ is at most the probability that $\mathbf{x} \sim N(\mathbf{c}_i', \sigma^2 \mathbb{I})$ is in $S$. This is immediate by a coupling of the two random variables: we let the first random variable be $\mathbf{c}_i + \mathbf{z}$ and the second random variable be $\mathbf{c}_i' + \mathbf{z}$, where the choice of $i$ and Gaussian noise $\mathbf{z}$ are the same for both random variables. For any monotonic $S$, since $\mathbf{c}_i' - \mathbf{c}_i$ is non-negative, $\mathbf{c}_i + \mathbf{z}$ is in $S$ only if $\mathbf{c}_i' + \mathbf{z}$ is in $S$, giving that the probability $\mathbf{x} \sim N(\mathbf{c}_i, \sigma^2 \mathbb{I})$ is in $S$ is at most the probability $\mathbf{x} \sim N(\mathbf{c}_i', \sigma^2 \mathbb{I})$ is in $S$. $\qquad\square$

Since the above proof holds for any $\mathbf{c}_i, \mathbf{c}_i'$ satisfying the assumptions in the lemma, it also holds if $\mathbf{c}_i'$ are fixed/non-adaptive but the entries in $\mathbf{c}_i$ are chosen adaptively (while still satisfying the assumptions in the lemma), i.e. the $j$th coordinate of $\mathbf{c}_i$ is chosen only after seeing the first $j-1$ coordinates of the output. In the scalar case, we get the following corollary:

**Corollary B.3.** *Let $\{p_1, p_2, \ldots p_k\}, \{c_1, c_2, \ldots, c_k\}$ and $\{p_1', p_2', \ldots p_{k'}'\}, \{c_1', c_2', \ldots c_{k'}'\}$ be such that for all $T$, $\sum_{i:c_i' \geq T} p_i' \geq \sum_{i:c_i \geq T} p_i$. In other words, the random variable induced by $\{p_i\}_i, \{c_i\}_i$ is stochastically dominated by the random variable induced by $\{p_i'\}_i, \{c_i'\}_i$. We also assume $c_i, c_i' \geq 0$ for all $i$.*

*Then the PLD of*

$$\mathcal{M}_{MoG}(\{p_1, p_2, \ldots p_k\}, \{c_1, c_2, \ldots, c_k\})$$

*is dominated by the PLD of*

$$\mathcal{M}_{MoG}(\{p_1', p_2', \ldots p_{k'}'\}, \{c_1', c_2', \ldots c_{k'}'\}).$$

Cor. B.3 follows from Lem. B.2 since by allowing duplicate $c_i$ values, we can reduce to the setting where the probabilities are the same, and $c_i \leq c_i'$ for all $c_i$. For example, if $c_i$ is 0 or 1 w.p. 1/2 and $c_i'$ is 0, 1, or 2 w.p. 1/3, we can use $\{p_i\} = \{1/3, 1/6, 1/6, 1/3\}$, $\{c_i\} = \{0, 0, 1, 1\}$, and $\{c_i'\} = \{0, 1, 1, 2\}$.

## B.2 DIMENSION REDUCTION FOR MOG MECHANISMS

We now give the following lemma, which lets us reduce the dimensions of a VMoG mechanism.

**Lemma B.4.** *Let $\mathbf{c}_1, \mathbf{c}_2, \ldots, \mathbf{c}_k \in \mathbb{R}^{n \times p}$. Let $\mathbf{c}_1', \mathbf{c}_2', \ldots, \mathbf{c}_k' \in \mathbb{R}^n$ be vectors such that $\|(\mathbf{c}_i)_{j,:}\|_2 \leq \mathbf{c}_i'(j)$ for all $i, j$, i.e. the entries of $\mathbf{c}_i'$ upper bound the $\ell_2$-norms of the corresponding rows of $\mathbf{c}_i$. Then the PLD of*

$$\mathcal{M}_{VMoG}(\{p_1, p_2, \ldots, p_k\}, \{\mathbf{c}_1, \mathbf{c}_2, \ldots, \mathbf{c}_k\})$$

*is dominated by the PLD of*

$$\mathcal{M}_{VMoG}(\{p_1, p_2, \ldots, p_k\}, \{\mathbf{c}_1', \mathbf{c}_2', \ldots, \mathbf{c}_k'\}).$$

*Furthermore, this holds even if the rows of each $\mathbf{c}_i$ are adaptively chosen and $\mathbf{c}_i'$ are fixed, i.e. the $j$th row of all $\mathbf{c}_i$ is chosen by an adversary after seeing the first $j-1$ rows of the output of the VMoG mechanism, as long as the assumption $\|(\mathbf{c}_i)_{j,:}\|_2 \leq \mathbf{c}_i'(j)$ holds.*

We need the following lemma, which we can apply multiple times to prove Lem. B.4:

**Lemma B.5.** *Let $w_1, w_2, \ldots w_k > 0$ be positive scalars and let $\mathbf{c}_1, \mathbf{c}_2, \ldots \mathbf{c}_k \in \mathbb{R}^p$ be arbitrary vectors. Then for any $\varepsilon$ and $\sigma > 0$:*

$$\mathbb{E}_{\mathbf{x} \sim N(\mathbf{0}, \sigma^2 \mathbb{I}_p)} \left[ \max \left\{ \sum_i w_i \exp(\langle \mathbf{c}_i, \mathbf{x} \rangle) - e^\varepsilon, 0 \right\} \right] \leq \mathbb{E}_{x \sim N(0, \sigma^2)} \left[ \max \left\{ \sum_i w_i \exp(\|\mathbf{c}_i\|_2 \, x) - e^\varepsilon, 0 \right\} \right].$$

*Proof.* $\sum_i w_i \exp(\|\mathbf{c}_i\|_2 \, x)$ as a function of $x$ is continuous, increasing in $x$, and has range $\mathbb{R}^+$. So, there exists some $t$ such that $\sum_i w_i \exp(\|\mathbf{c}_i\|_2 \, t) = e^\varepsilon$. For this choice of $t$, let $t_i = w_i \exp(\|\mathbf{c}_i\|_2 \, t)$. Then we have for all $x$:

$$\max \left\{ \sum_i w_i \exp(\|\mathbf{c}_i\|_2 \, x) - e^\varepsilon, 0 \right\} = \sum_i \max \left\{ w_i \exp(\|\mathbf{c}_i\|_2 \, x) - t_i, 0 \right\}.$$

Now, by linearity of expectation and the fact that $\max\{\sum_i a_i, \sum_i b_i\} \leq \sum_i \max\{a_i, b_i\}$:

$$\mathbb{E}_{\mathbf{x} \sim N(\mathbf{0}, \sigma^2 \mathbb{I}_p)} \left[ \max \left\{ \sum_i w_i \exp(\langle \mathbf{c}_i, \mathbf{x} \rangle) - e^\varepsilon, 0 \right\} \right] \leq \mathbb{E}_{\mathbf{x} \sim N(\mathbf{0}, \sigma^2 \mathbb{I}_p)} \left[ \sum_i \max \left\{ w_i \exp(\langle \mathbf{c}_i, \mathbf{x} \rangle) - t_i, 0 \right\} \right]$$

$$= \sum_i \mathbb{E}_{\mathbf{x} \sim N(\mathbf{0}, \sigma^2 \mathbb{I}_p)} \left[ \max \left\{ w_i \exp(\langle \mathbf{c}_i, \mathbf{x} \rangle) - t_i, 0 \right\} \right]$$

$$= \sum_i \mathbb{E}_{x \sim N(0, \sigma^2)} \left[ \max \left\{ w_i \exp(\|\mathbf{c}_i\|_2 \, x) - t_i, 0 \right\} \right]$$

$$= \mathbb{E}_{x \sim N(0, \sigma^2)} \left[ \sum_i \max \left\{ w_i \exp(\|\mathbf{c}_i\|_2 \, x) - t_i, 0 \right\} \right]$$

$$= \mathbb{E}_{x \sim N(0, \sigma^2)} \left[ \max \left\{ \sum_i w_i \exp(\|\mathbf{c}_i\|_2 \, x) - e^\varepsilon, 0 \right\} \right].$$

$\square$

*Proof of Lem. B.4.* Lem. B.2 holds for adaptively chosen $\mathbf{c}_i$ and fixed $\mathbf{c}'_i$ (using the notation of that lemma), so by Lem. B.2 it suffices to prove the lemma for adaptive $\mathbf{c}_i$ and fixed $\mathbf{c}'_i$ such that $\|(\mathbf{c}_i)_{j,:}\|_2 = \mathbf{c}'_i(j)$ for all $i, j$. Further, by Lem. B.1, it suffices to show the lemma under the remove adjacency. That is, $P = N(\mathbf{c}_i, \sigma^2 \mathbb{I}_{n \times p})$, $Q = N(\mathbf{0}, \sigma^2 \mathbb{I}_{n \times p})$, $P' = N(\mathbf{c}'_i, \sigma^2 \mathbb{I}_n)$, $Q' = N(\mathbf{0}, \sigma^2 \mathbb{I}_n)$, and it suffices to show $H_\varepsilon(P, Q) \leq H_\varepsilon(P', Q')$ for all $\varepsilon$.

We have:

$$H_\varepsilon(P, Q) = \mathbb{E}_{\mathbf{x} \sim Q} \left[ \max \left\{ \frac{P(x)}{Q(x)} - e^\varepsilon, 0 \right\} \right]$$

$$= \mathbb{E}_{\mathbf{x} \sim N(\mathbf{0}, \sigma^2 \mathbb{I}_{n \times p})} \left[ \max \left\{ \frac{\sum_i p_i \exp(\|\mathbf{x} - \mathbf{c}_i\|_2^2 / 2\sigma^2)}{\exp(\|\mathbf{x}\|_2^2 / 2\sigma^2)} - e^\varepsilon, 0 \right\} \right]$$

$$= \mathbb{E}_{\mathbf{x} \sim N(\mathbf{0}, \sigma^2 \mathbb{I}_{n \times p})} \left[ \max \left\{ \sum_i p_i \exp \left( \frac{2 \langle \mathbf{c}_i, \mathbf{x} \rangle - \|\mathbf{c}_i\|_2^2}{2\sigma^2} \right) - e^\varepsilon, 0 \right\} \right]$$

To reflect the fact that $\mathbf{c}_i$ can be chosen adaptively, let $\mathbf{c}_{i,j}(\mathbf{x}_{1:j-1})$ denote any adversary's adaptive choice of the jth row of $\mathbf{c}_i$ after observing the first $j-1$ rows of $\mathbf{x}$. We can then write $H_\varepsilon(P,Q)$ as:

$$\mathbb{E}_{\mathbf{x}_1,\mathbf{x}_2,\ldots\mathbf{x}_n \sim N(\mathbf{0},\sigma^2\mathbb{I}_p)}\left[\max\left\{\sum_i p_i \prod_{j\in[n]} \exp\left(\frac{2\langle\mathbf{c}_{i,j}(\mathbf{x}_{1:j-1}),\mathbf{x}_j\rangle - \|\mathbf{c}_{i,j}(\mathbf{x}_{1:j-1})\|_2^2}{2\sigma^2}\right) - e^\varepsilon, 0\right\}\right] =$$

$$\mathbb{E}_{\mathbf{x}_1,\mathbf{x}_2,\ldots\mathbf{x}_{n-1} \sim N(\mathbf{0},\sigma^2\mathbb{I}_p)}\left[\mathbb{E}_{\mathbf{x}_n \sim N(\mathbf{0},\sigma^2\mathbb{I}_p)}\left[\max\left\{\sum_i p_i \prod_{j\in[n]} \exp\left(\frac{2\langle\mathbf{c}_{i,j}(\mathbf{x}_{1:j-1}),\mathbf{x}_j\rangle - \|\mathbf{c}_{i,j}(\mathbf{x}_{1:j-1})\|_2^2}{2\sigma^2}\right) - e^\varepsilon, 0\right\}\right]\right].$$

$$(1)$$

Note that the values of all $\mathbf{c}_{i,j}$ in (1) are constants with respect to the inner expectation. So for any realization of $\mathbf{x}_1, \mathbf{x}_2, \ldots, \mathbf{x}_{n-1}$, choosing

$$w_i = p_i \exp\left(-\frac{\|\mathbf{c}_{i,n}(\mathbf{x}_{1:n-1})\|_2^2}{2\sigma^2}\right)\prod_{j\in[n-1]}\exp\left(\frac{2\langle\mathbf{c}_{i,j}(\mathbf{x}_{1:j-1}),\mathbf{x}_j\rangle - \|\mathbf{c}_{i,j}(\mathbf{x}_{1:j-1})\|_2^2}{2\sigma^2}\right)$$

and observing that by assumption $\|\mathbf{c}_{i,n}(\mathbf{x}_{1:n-1})\|_2 = \mathbf{c}'_i(n)$, we can apply Lem. B.5 to upper bound the inner expectation in (1) as:

$$(1) \le \mathbb{E}_{\mathbf{x}_1,\mathbf{x}_2,\ldots\mathbf{x}_{n-1} \sim N(\mathbf{0},\sigma^2\mathbb{I}_p), x_n \sim N(0,\sigma^2)}\left[\max\left\{\sum_i p_i \prod_{j\in[n-1]}\right.\right.$$

$$\left.\left.\cdot \exp\left(\frac{2\langle\mathbf{c}_{i,j}(\mathbf{x}_{1:j-1}),\mathbf{x}_j\rangle - \|\mathbf{c}_{i,j}(\mathbf{x}_{1:j-1})\|_2^2}{2\sigma^2}\right) \cdot \exp\left(\frac{2\mathbf{c}'_i(n)x_n - \mathbf{c}'_i(n)^2}{2\sigma^2}\right) - e^\varepsilon, 0\right\}\right].$$

We can then iteratively repeat this argument for rows $n-1$, $n-2$, $\ldots 1$ to get:

$$H_\varepsilon(P,Q) \le \mathbb{E}_{\mathbf{x}_1,\mathbf{x}_2,\ldots\mathbf{x}_{n-1} \sim N(\mathbf{0},\sigma^2\mathbb{I}_p), x_n \sim N(0,\sigma^2)}\left[\max\left\{\sum_i p_i\right.\right.$$

$$\left.\left.\cdot\prod_{j\in[n-1]}\exp\left(\frac{2\langle\mathbf{c}_{i,j}(\mathbf{x}_{1:j-1}),\mathbf{x}_j\rangle - \|\mathbf{c}_{i,j}(\mathbf{x}_{1:j-1})\|_2^2}{2\sigma^2}\right) \cdot \exp\left(\frac{2\mathbf{c}'_i(n)x_n - \mathbf{c}'_i(n)^2}{2\sigma^2}\right) - e^\varepsilon, 0\right\}\right]$$

$$\le \mathbb{E}_{\mathbf{x}_1,\mathbf{x}_2,\ldots\mathbf{x}_{n-2} \sim N(\mathbf{0},\sigma^2\mathbb{I}_p), x_{n-1},x_n \sim N(0,\sigma^2)}\left[\max\left\{\sum_i p_i\right.\right.$$

$$\left.\left.\prod_{j\in[n-2]}\exp\left(\frac{2\langle\mathbf{c}_{i,j}(\mathbf{x}_{1:j-1}),\mathbf{x}_j\rangle - \|\mathbf{c}_{i,j}(\mathbf{x}_{1:j-1})\|_2^2}{2\sigma^2}\right) \cdot \prod_{j\in[n]\setminus[n-2]}\exp\left(\frac{2\mathbf{c}'_i(j)x_j - \mathbf{c}'_i(j)^2}{2\sigma^2}\right) - e^\varepsilon, 0\right\}\right]$$

$$\le \mathbb{E}_{\mathbf{x}_1,\mathbf{x}_2,\ldots\mathbf{x}_{n-3} \sim N(\mathbf{0},\sigma^2\mathbb{I}_p), x_{n-2},x_{n-1},x_n \sim N(0,\sigma^2)}\left[\max\left\{\sum_i p_i\right.\right.$$

$$\left.\left.\prod_{j\in[n-3]}\exp\left(\frac{2\langle\mathbf{c}_{i,j}(\mathbf{x}_{1:j-1}),\mathbf{x}_j\rangle - \|\mathbf{c}_{i,j}(\mathbf{x}_{1:j-1})\|_2^2}{2\sigma^2}\right) \cdot \prod_{j\in[n]\setminus[n-3]}\exp\left(\frac{2\mathbf{c}'_i(j)x_j - \mathbf{c}'_i(j)^2}{2\sigma^2}\right) - e^\varepsilon, 0\right\}\right]$$

$$\cdots$$

$$\leq \mathbb{E}_{x_1,x_2,\ldots,x_n \sim N(0,\sigma^2)} \left[ \max \left\{ \sum_i p_i \prod_{j \in [n]} \exp\left( \frac{2\mathbf{c}_i'(j)x_j - \mathbf{c}_i'(j)^2}{2\sigma^2} \right) - e^\varepsilon, 0 \right\} \right]$$

$$= \mathbb{E}_{\mathbf{x} \sim N(0,\sigma^2 \mathbb{I}_n)} \left[ \max \left\{ \sum_i p_i \exp\left( \frac{2\langle \mathbf{c}_i', \mathbf{x}\rangle - \|\mathbf{c}_i'\|_2^2}{2\sigma^2} \right) - e^\varepsilon, 0 \right\} \right] = H_\varepsilon(P', Q').$$

$\square$

As a corollary to the above "matrix-to-vector reduction", we have a "vector-to-scalar reduction" for MoG mechanisms:

**Corollary B.6.** *The PLD of*

$$\mathcal{M}_{VMoG}(\{p_1, p_2, \ldots, p_k\}, \{\mathbf{c}_1, \mathbf{c}_2, \ldots, \mathbf{c}_k\})$$

*is dominated by the PLD of*

$$\mathcal{M}_{MoG}(\{p_1, p_2, \ldots, p_k\}, \{\|\mathbf{c}_1\|_2, \|\mathbf{c}_2\|_2, \ldots, \|\mathbf{c}_k\|_2\}).$$

## C  DEFERRED PROOFS

### C.1  PROOF OF THEOREM C.2

We will need the following lemma, which shows domination is preserved by composition:

**Lemma C.1** (Theorem 10 in (Zhu et al., 2022)). *Let $\mathcal{M}_1, \ldots, \mathcal{M}_k$ be an adaptive sequence of mechanisms, i.e., each mechanism receives the output of all previous mechanism and the database. Suppose for all $i$ and joint outputs $x$ of $\mathcal{M}_1, \ldots \mathcal{M}_{i-1}$, the PLD of $\mathcal{M}_i(x, D)$ and $\mathcal{M}_i(x, D')$ is dominated by the PLD of $P_i, Q_i$. Then letting $\mathcal{M}$ be the composition of these mechanisms, the PLD of $\mathcal{M}(D), \mathcal{M}(D')$ is dominated by the PLD of $P_1 \times P_2 \times \ldots, Q_1 \times Q_2 \times \ldots$.*

*Similarly, if $L_1, L_2, \ldots, L_k$ and $L_1', L_2', \ldots, L_k'$ are random variables such that $L_i$ dominates $L_i'$ for all $i$, then $L_1 + L_2 + \ldots + L_k$ dominates $L_1' + L_2' + \ldots + L_k'$.*

In (Zhu et al., 2022), only the first part of Lem. C.1 is stated. However, the proof allows arbitrary measures, i.e., measures that don't integrate to 1, which implies the second part of Lem. C.1.

**Theorem C.2.** *Let $\mathcal{M}_1 : \mathcal{D} \to \mathcal{X}_1, \mathcal{M}_2 : \mathcal{X}_1 \times \mathcal{D} \to \mathcal{X}_2, \mathcal{M}_3 : \mathcal{X}_1 \times \mathcal{X}_2 \times \mathcal{D} \to \mathcal{X}_3, \ldots \mathcal{M}_n$ be a sequence of adaptive mechanisms, where each $\mathcal{M}_i$ takes a dataset in $\mathcal{D}$ and the output of mechanisms $\mathcal{M}_1, \ldots, \mathcal{M}_{i-1}$ as input. Let $\mathcal{M}$ be the mechanism that outputs $(x_1 = \mathcal{M}_1(D), x_2 = \mathcal{M}_2(x_1, D), \ldots, x_n = \mathcal{M}_n(x_1, \ldots, x_{n-1}, D))$. Fix any two adjacent datasets $D, D'$.*

*Suppose there exists "bad events" $E_1 \subseteq \mathcal{X}_1, E_2 \subseteq \mathcal{X}_1 \times \mathcal{X}_2, \ldots \ldots E_{n-1} \subseteq \mathcal{X}_1 \times \mathcal{X}_2 \times \ldots \times \mathcal{X}_{n-1}$ such that*

$$\Pr_{x \sim \mathcal{M}(D)} [\exists i : (x_1, x_2, \ldots x_i) \in E_i] \leq \delta$$

*and pairs of distributions $(P_1, Q_1), (P_2, Q_2), \ldots (P_n, Q_n)$ such that the PLD of $\mathcal{M}_1(D)$ and $\mathcal{M}_1(D')$ is dominated by the PLD of $P_1, Q_1$ and for any $i \geq 1$ and "good" output $(x_1, x_2, \ldots x_i) \notin E_i$, the PLD of $\mathcal{M}_{i+1}(x_1, \ldots, x_i, D)$ and $\mathcal{M}_{i+1}(x_1, \ldots, x_i, D')$ is dominated by the PLD of $P_{i+1}, Q_{i+1}$. Then for all $\varepsilon$:*

$$H_\varepsilon(\mathcal{M}(D), \mathcal{M}(D')) \leq H_\varepsilon(P_1 \times P_2 \times \ldots \times P_n, Q_1 \times Q_2 \times \ldots \times Q_n) + \delta.$$

---

**Algorithm 2** `ProbabilityTailBounds`$(\mathbf{C}, p, \sigma, \delta_1)$

---

1: **Input:** Matrix $\mathbf{C}$, sampling probability $p$, noise standard deviation $\sigma$, probability $\delta_1$.
2: $\delta' = \frac{\delta_1}{2 \cdot (nnz(\mathbf{C}) - n)}$ $\qquad\qquad\qquad\qquad\qquad\qquad$ ▷ $nnz$ is the number of non-zeros.
3: $z = \Phi^{-1}(1 - \delta')$ $\qquad\qquad$ ▷ Tail bound on normal distribution; here, $\Phi$ is the standard normal CDF.
4: **for** $i, j \in [n]$ **do**
5: $\quad$ **if** $\mathbf{C}_{i,j} = 0$ **then**
6: $\qquad$ $\widetilde{p}_{i,j} = 1$
7: $\quad$ **else**
8: $\qquad$ $s_{i,j} = $ minimum $s$ s.t. $\mathbf{Pr}[\sum_{j' \leq i} x_{j'} \langle \mathbf{C}_{1:i-1,j}, \mathbf{C}_{1:i-1,j'} \rangle > s] \leq \delta', x_{j'} \overset{i.i.d.}{\sim} Bern(p)$
$\qquad$ ▷ $s_{i,j}$ is a tail bound on the dot product of first $i - 1$ entries of $\mathbf{C}x$ and $\mathbf{C}_{1:i-1,j}$.
9: $\qquad$ $\varepsilon_{i,j} = \frac{z\|\mathbf{C}_{1:i-1,j}\|_2}{\sigma} + \frac{2s_{i,j} - \|\mathbf{C}_{1:i-1,j}\|_2^2}{2\sigma^2}$
$\qquad$ ▷ $\varepsilon_{i,j}$ is a tail bound on the privacy loss of a participation in round $j$ after outputting first $i - 1$ rounds
10: $\qquad$ $\widetilde{p}_{i,j} = \frac{p \cdot \exp(\varepsilon_{i,j})}{p \cdot \exp(\varepsilon_{i,j}) + (1 - p)}$
11: **return** $\{\widetilde{p}_{i,j}\}_{i,j \in [n]}$.

---

Figure 5: Algorithm for computing $\widetilde{p}_{i,j}$, tail bound on conditional probability of participating in round $j$ given first $i - 1$ outputs.

*Proof.* Let $L_1$ be the privacy loss random variable of $\mathcal{M}$, and let $L_2$ be the privacy loss random variable of $P_1 \times P_2 \times \ldots \times P_n, Q_1 \times Q_2 \times \ldots \times Q_n$. We want to show $H_\varepsilon(L_1) \leq H_\varepsilon(L_2) + \delta$ for all $\delta$.

Let $L_1'$ be the random variable coupled with $L_1$, with the coupling defined as follows: If $\exists i : (x_1, x_2, \ldots, x_i) \in E_i$, then $L_1' = -\infty$, otherwise $L_1' = L_1$. Let $E = \{x | \exists i : (x_1, x_2, \ldots x_i) \in E_i\}$. Then for all $\varepsilon$:

$$H_\varepsilon(L_1) = \mathbb{E}_x\left[\max\left\{1 - e^{\varepsilon - L_1(x)}, 0\right\}\right]$$

$$= \mathbf{Pr}_x[x \notin E] \cdot \mathbb{E}_x\left[\max\left\{1 - e^{\varepsilon - L_1(x)}, 0\right\}\Big| x \notin E\right] + \mathbf{Pr}_x[x \in E] \cdot \mathbb{E}_x\left[\max\left\{1 - e^{\varepsilon - L_1(x)}, 0\right\}\Big| x \in E\right]$$

$$= H_\varepsilon(L_1') + \mathbf{Pr}_x[x \in E] \cdot \mathbb{E}_x\left[\max\left\{1 - e^{\varepsilon - L_1(x)}, 0\right\}\Big| x \in E\right] \leq H_\varepsilon(L_1') + \mathbf{Pr}_x[x \in E] \leq H_\varepsilon(L_1') + \delta.$$

So it suffices to show $L_1'$ is dominated by $L_2$. We consider the following process for sampling $L_1'$: For each $i$, if for any $i' < i$, $(x_1, x_2, \ldots, x_{i'}) \in E_{i'}$, then we let $L_{1,i} = -\infty$ deterministically. Otherwise we sample $x_i \sim \mathcal{M}_i(x_1, \ldots, x_{i-1}, D)$, $L_{1,i} = \ln\left(\frac{\mathbf{Pr}_{y_i \sim \mathcal{M}_i(x_1, \ldots, x_{i-1}, D)}[y_i = x_i]}{\mathbf{Pr}_{y_i \sim \mathcal{M}_i(x_1, \ldots, x_{i-1}, D')}[y_i = x_i]}\right)$. Then $L_1' = \sum_i L_{1,i}'$. Similarly, let $L_{2,i}$ be the privacy loss random variable for $P_i, Q_i$, and let $L_2 = \sum_i L_{2,i}$. By assumption, the distribution of $L_{1,i}'$ conditioned on $x_1, x_2, \ldots, x_{i-1}$ is always dominated by $L_{2,i}$. So by Lem. C.1, $L_1'$ is dominated by $L_2$. $\qquad\square$

## C.2 PROOF OF THM. 4.3

Before stating the proof, in Fig. 5 we give `ProbabilityTailBounds`, the subroutine used to compute the values of $\widetilde{p}_{i,j}$.

*Proof of Thm. 4.3.* For simplicity in the proof we only consider remove adjacency, i.e. $D$ contains a sensitive example zeroed out in $D'$. By symmetry the proof also works for add adjacency. By quasi-convexity

of approximate DP, it suffices to prove the theorem assuming the participation of all examples except the sensitive example is deterministic, i.e. we know the contribution of all examples except the sensitive example to $\mathbf{x}$, so we can assume these contributions are zero. So, let $\mathbf{x}$ be the matrix used in the matrix mechanism if we were to sample the sensitive example in each round. Then, the matrix mechanism is a VMoG mechanism with probabilities $\{p^{|S|}(1-p)^{n-|S|}\}_{S \subseteq [n]}$ and sensitivities $\{\sum_{j \in S} \mathbf{C}_{:,j}\mathbf{x}_j\}_{S \subseteq [n]}$.

Our proof proceeds in three high-level steps:

1. We show the matrix mechanism is dominated by a sequence of adaptively chosen MoG mechanisms.
2. We show each of the adaptively chosen MoG mechanisms is further dominated by a PMoG mechanism.
3. We show these PMoG mechanisms are with high probability dominated by the PMoG mechanisms in MMCC, and then apply Theorem C.2.

**Step 1 (matrix mechanism dominated by sequence of MoG mechanisms):** Let $f$ be the function that takes a matrix $\mathbf{M}$ and returns a vector $f(\mathbf{M})$ where the $i$th entry of this vector is the $\ell_2$-norm of the $i$th row of $\mathbf{M}$. Using triangle inequality, for any $\mathbf{x}$ such that each row of $\mathbf{x}$ has norm at most 1, $f(\sum_{j \in S} \mathbf{C}_{:,j}\mathbf{x}_j)$ is entrywise less than or equal to $\sum_{j \in S} \mathbf{C}_{:,j}$. So by Lem. B.4 the matrix mechanism is dominated by the VMoG mechanism with probabilities $\{p^{|S|}(1-p)^{n-|S|}\}_{S \subseteq [n]}$ and sensitivities $\{\sum_{j \in S} \mathbf{C}_{:,j}\}_{S \subseteq [n]}$[7]. Note that this is exactly the (non-adaptive) matrix mechanism where each $\mathbf{x}_i = 1$ (prior to sampling), i.e. it suffices to prove the privacy guarantee holds for this choice of $\mathbf{x}$. So, for the rest of the proof we will assume the input of the matrix mechanism (prior to sampling) is the all ones vector.

Now, let $\theta_{1:i}$ denote the output of rounds 1 to $i$. By Observation 3.2, this random variable is the same as the composition over $i$ of outputting $\theta_i$ sampled from its distribution conditioned on $\theta_{1:i-1}$. Let $S_i$ denote the set of rounds in $[i]$ in which we sample the sensitive example. Abusing notation to let $\mathbf{Pr}$ denote a likelihood, the likelihood of the matrix mechanism $\mathcal{M}(D)$ outputting $\theta_i$ in round $i$ conditioned on $\theta_{1:i-1}$ is:

$$\sum_{T \subseteq [i]} \mathbf{Pr}_{\tau \sim \Theta(D)}[S_i = T | \tau_{1:i-1} = \theta_{1:i-1}] \mathbf{Pr}_{\tau_i \sim N(\sum_{j \in T} \mathbf{C}_{i,j}, \sigma^2 \cdot \mathbb{I})}[\tau_i = \theta_i]$$

The likelihood of $\mathcal{M}(D')$ outputting $\theta_i$ in round $i$ (conditioned on $\theta_{1:i-1}$, which doesn't affect the likelihood since since each coordinate of $\theta$ is independent when sampled from $\mathcal{M}(D')$) is

$$\mathbf{Pr}_{\tau_i \sim N(\mathbf{0}, \sigma^2 \cdot \mathbb{I})}[\tau_i = \theta_i].$$

In other words, the distribution of $\theta_i$ conditioned on $\theta_{1:i-1}$ under $\mathcal{M}(D), \mathcal{M}(D')$ is exactly the same as the pairs of output distributions given by the MoG mechanism.

$$\mathcal{M}_{MoG}\left(\left\{\mathbf{Pr}_{\tau \sim \Theta(D)}[S_i = T | \tau_{1:i-1} = \theta_{1:i-1}]\right\}_{T \subseteq [i]}, \{\sum_{j \in T} \mathbf{C}_{i,j}\}_{T \subseteq [i]}\right).$$

So the matrix mechanism with $\mathbf{x}$ being all ones is the same as the sequence of (adaptively chosen) MoG mechanisms given by

---

[7]Note that since $\mathbf{C}$ is lower-triangular, so the choice of the distribution of the $i$th row of $\mathbf{Cx}$ by an adaptive adversary depends only on rows 1 to $i-1$ of $\mathbf{Cx} + \mathbf{z}$. That is, an adversary who chooses the $j$th row of $\mathbf{x}$ after seeing the $j-1$st first rows of the matrix mechanism satisfies the adaptivity condition in Lem. B.4.

$$\left\{ \mathcal{M}_{MoG} \left( \left\{ \mathbf{Pr}_{\tau \sim \Theta(D)}[S_i = T | \tau_{1:i-1} = \theta_{1:i-1}] \right\}_{T \subseteq [i]}, \{ \sum_{j \in T} \mathbf{C}_{i,j} \}_{T \subseteq [i]} \right) \right\}_{i \in [n]}.$$

**Step 2 (each MoG is dominated by a PMoG):** To achieve step 2, we use the following lemma:

**Lemma C.3.** *Let*

$$p_{i,j} = \frac{p \exp \left( \frac{2 \langle \theta_{1:i-1}, \mathbf{C}_{1:i-1,j} \rangle - \| \mathbf{C}_{1:i-1,j} \|_2^2}{2\sigma^2} \right)}{p \exp \left( \frac{2 \langle \theta_{1:i-1}, \mathbf{C}_{1:i-1,j} \rangle - \| \mathbf{C}_{1:i-1,j} \|_2^2}{2\sigma^2} \right) + 1 - p}.$$

*The random variable induced by probabilities $\left\{ \prod_{j \in T} p_{i,j} \prod_{j \in [i] \setminus T} (1 - p_{i,j}) \right\}_{T \subseteq [i]}$ and support $\{ \sum_{j \in T} \mathbf{C}_{i,j} \}_{T \subseteq [i]}$ stochastically dominates the random variable induced by probabilities $\{ \mathbf{Pr}_{\tau \sim \Theta(D)}[S_i = T | \tau_{1:i-1} = \theta_{1:i-1}] \}_{T \subseteq [i]}$ and the same support.*

Proving Lem. C.3 completes the step as with this lemma and Cor. B.3, the PLD of

$$\mathcal{M}_{MoG} \left( \left\{ \mathbf{Pr}_{\tau \sim \Theta(D)}[S_i = T | \tau_{1:i-1} = \theta_{1:i-1}] \right\}_{T \subseteq [i]}, \{ \sum_{j \in T} \mathbf{C}_{i,j} \}_{T \subseteq [i]} \right).$$

is dominated by the PLD of

$$\mathcal{M}_{PMoG} \left( \{ p_{i,j} \}_{j \in [n]}, \{ \mathbf{C}_{i,j} \}_{j \in [n]} \right).$$

*Proof of Lem. C.3.* Sampling $T$ according to probabilities $\{ \mathbf{Pr}_{\tau \sim \Theta(D)}[S_i = T | \tau_{1:i-1} = \theta_{1:i-1}] \}_{T \subseteq [i]}$ is equivalent to the following process: We start with $T = \emptyset$, and for each $j \in [i]$, add it to $T$ with probability $\mathbf{Pr}[T \cup \{j\} \subseteq S_i | T \subseteq S_i, \tau_{1:i-1} = \theta_{1:i-1}]$. Similarly, sampling $T$ according to $\left\{ \prod_{j \in T} p_{i,j} \prod_{j \in [i] \setminus T} (1 - p_{i,j}) \right\}_{T \subseteq [i]}$ is equivalent to the same process, except we add $j$ with probability $p_{i,j}$. If we show that $\mathbf{Pr}[T \cup \{j\} \subseteq S_i | T \subseteq S_i, \tau_{1:i-1} = \theta_{1:i-1}] \leq p_{i,j}$ for all $T, j$, then we can couple these sampling processes such that with probability 1, $\sum_{j \in T} \mathbf{C}_{i,j}$ is at least as large for the second process as for the first, which implies the lemma. The posterior distribution of $S_i$ satisfies:

$$\mathbf{Pr}_{\tau \sim \Theta(D)}[S_i = T | \tau_{1:i-1} = \theta_{1:i-1}] \propto \mathbf{Pr}_{\tau \sim \Theta(D)}[S_i = T] \cdot \mathbf{Pr}_{\tau \sim \Theta(D)}[\tau_{1:i-1} = \theta_{1:i-1} | S_i = T]$$

$$\propto p^{|T|}(1 - p)^{i - |T|} \cdot \exp \left( \frac{2 \langle \theta_{1:i-1}, \sum_{j \in T} \mathbf{C}_{1:i-1,j} \rangle - \left\| \sum_{j \in T} \mathbf{C}_{1:i-1,j} \right\|_2^2}{2\sigma^2} \right).$$

Hence:

$$\mathbf{Pr}[T \cup \{j\} \subseteq S_i | T \subseteq S_i, \tau_{1:i-1} = \theta_{1:i-1}] =$$

$$\frac{\sum_{T' \supseteq T \cup \{j\}} p^{|T'|}(1-p)^{i-|T'|} \cdot \exp\left(\frac{2\langle\theta_{1:i-1}, \sum_{j\in T'} \mathbf{C}_{1:i-1,j}\rangle - \left\|\sum_{j'\in T'} \mathbf{C}_{1:i-1,j'}\right\|_2^2}{2\sigma^2}\right)}{\sum_{T' \supseteq T} p^{|T'|}(1-p)^{i-|T'|} \cdot \exp\left(\frac{2\langle\theta_{1:i-1}, \sum_{j\in T'} \mathbf{C}_{1:i-1,j}\rangle - \left\|\sum_{j'\in T'} \mathbf{C}_{1:i-1,j'}\right\|_2^2}{2\sigma^2}\right)}.$$

Fix some $T' \supseteq T \cup \{j\}$. Consider the term in the numerator sum corresponding to $T'$, and the two terms in the denominator sum corresponding to $T'$ and $T' \setminus \{j\}$. The ratio of the numerator term to the sum of the two denominator terms is:

$$\frac{p \cdot \exp\left(\frac{2\langle\theta_{1:i-1}, \mathbf{C}_{1:i-1,j}\rangle - \left\|\sum_{j'\in T'} \mathbf{C}_{1:i-1,j'}\right\|_2^2}{2\sigma^2}\right)}{p \cdot \exp\left(\frac{2\langle\theta_{1:i-1}, \mathbf{C}_{1:i-1,j}\rangle - \left\|\sum_{j'\in T'} \mathbf{C}_{1:i-1,j'}\right\|_2^2}{2\sigma^2}\right) + (1-p) \cdot \exp\left(\frac{-\left\|\sum_{j'\in T'\setminus\{j\}} \mathbf{C}_{1:i-1,j'}\right\|_2^2}{2\sigma^2}\right)}.$$

Since entries of $\mathbf{C}$ are non-negative, we have $\left\|\sum_{j'\in T'} \mathbf{C}_{1:i-1,j'}\right\|_2^2 \geq \left\|\sum_{j'\in T'\setminus j} \mathbf{C}_{1:i-1,j'}\right\|_2^2 + \|\mathbf{C}_{1:i-1,j'}\|_2^2$, hence this ratio and thus $\mathbf{Pr}[T \cup \{j\} \subseteq S_i | T \subseteq S_i, \tau_{1:i-1} = \theta_{1:i-1}]$ are at most $p_{i,j}$, which proves the lemma. $\square$

**Step 3 (replacing $p_{i,j}$ with $\widetilde{p}_{i,j}$ via conditional composition):** By Theorem C.2 and Cor. B.3, it now suffices to show that w.p. $1 - \delta_1$, $p_{i,j} \leq \widetilde{p}_{i,j}$ for all $i, j$ simultaneously. The bound trivially holds for entries where $\mathbf{C}_{i,j} = 0$, so we only need the bound to hold for all $nnz(\mathbf{C})$ pairs $i, j$ such that $\mathbf{C}_{i,j} > 0$. Furthermore, if $\mathbf{C}_{i,j}$ is the first non-zero entry of column $j$, then $\mathbf{C}_{1:i-1,j}$ is the all zero-vector, so we get $p_{i,j} = \widetilde{p}_{i,j} = p$.

So, there are only $nnz(\mathbf{C}) - n$ "non-trivial" pairs we need to prove the tail bound for; by a union bound, we can show each of these bounds individually holds w.p. $\frac{\delta_1}{nnz(\mathbf{C})-n}$. By definition of $p_{i,j}, \widetilde{p}_{i,j}$, this is equivalent to showing $\langle\theta_{1:i-1}, \mathbf{C}_{1:i-1,j}\rangle \leq z \|\mathbf{C}_{1:i-1,j}\|_2 \sigma + s_{i,j}$ for each of these $i, j$ pairs. We have:

$$\langle\theta_{1:i-1}, \mathbf{C}_{1:i-1,j}\rangle = \sum_{j'\in S_i} \langle\mathbf{C}_{1:i-1,j'}, \mathbf{C}_{1:i-1,j}\rangle + \langle\mathbf{z}_{1:i-1}, \mathbf{C}_{1:i-1,j}\rangle.$$

The first term is tail bounded by $s_{i,j}$ with probability $1 - \frac{\delta_1}{2(nnz(\mathbf{C})-n)}$ by definition, the second term is drawn from $N(0, \|\mathbf{C}_{1:i-1,j}\|_2^2 \sigma^2)$ and thus tail bounded by $z \|\mathbf{C}_{1:i-1,j}\|_2 \sigma$ with the same probability by definition. A union bound over these two events gives the desired tail bound on $\langle\theta_{1:i-1}, \mathbf{C}_{1:i-1,j}\rangle$. $\square$

## C.3 PROOF OF LEM. 5.1

*Proof.* Since each $0 \leq p_i \leq 1/n'$, the mechanism is the same as the following: For each example we choose a subset $S \subseteq [n]$ of size $n'$ according to some distribution that is a function of the $p_i$, and then choose $i$ uniformly at random from the elements of $S$, and include the example in the $i$th subset. By quasi-convexity of approximate DP, it suffices to prove the DP guarantee for a fixed choice of $S$. For any fixed choice of $S$, the mechanism is equivalent to the shuffled Gaussian mechanism over $n'$ coordinates. Each unshuffled Gaussian mechanism satisfies $(\frac{\sqrt{2\ln(1.25/\delta_0)}}{\sigma}, \delta_0)$-DP, and then the lemma follows by the amplification via shuffling statement of Theorem 3.8 of (Feldman et al., 2022). $\square$

## C.4 PROOF OF THEOREM 5.2

We first analyze a simplified case where $\mathbf{x}_i = 0$ if $i \neq i^*$, and otherwise $\mathbf{x}_i = 1$ for $D$ and $\mathbf{x}_i = 0$ for $D'$. We later give a proof for the general case.

*Proof of Theorem 5.2 in simplified case.* Let $\tau_{j,k}$ be the value of the noisy sum $\sum_{k \cdot 2^j + 1 \leq i \leq (k+1) \cdot 2^j} \mathbf{x}_i + \mathbf{z}_{j,k}$, $\tau = \{\tau_{j,k}\}_{0 \leq j \leq \log n, 0 \leq k < n/2^j}$ and let $\Theta(D)$ be the distribution of these values under dataset $D$. We consider a single sensitive example; let $i^*$ be the (random) coordinate of $\mathbf{x}_{i^*}$ that this example contributes to.

Now, again abusing notation to let $\mathbf{Pr}$ denote a likelihood, we have for any $j$:

$$\Pr_{\tau \sim \Theta(D)} \left[ \{\tau_{j,k}\}_{0 \leq k < n/2^j} = \{\theta_{j',k}\}_{j'>j, 0 \leq k < n/2^{j'}} \Big| \{\tau_{j',k}\}_{j'>j, 0 \leq k < n/2^{j'}} = \{\theta_{j',k}\}_{j'>j, 0 \leq k < n/2^{j'}} \right] \propto$$

$$\sum_{0 \leq k^* \leq n/2^j} \Pr_{\tau \sim \Theta(D)} \left[ \{\tau_{j,k}\}_{0 \leq k < n/2^j} = \{\theta_{j',k}\}_{j'>j, 0 \leq k < n/2^{j'}} \Big| k^* \cdot 2^j + 1 \leq i^* \leq (k^*+1) \cdot 2^j \right] \cdot$$

$$\Pr_{\tau \sim \Theta(D)} \left[ k^* \cdot 2^j + 1 \leq i^* \leq (k^*+1) \cdot 2^j \Big| \{\tau_{j',k}\}_{j'>j, 0 \leq k < n/2^{j'}} = \{\theta_{j',k}\}_{j'>j, 0 \leq k < n/2^{j'}} \right]$$

In other words, for any $j$, the distribution of the $j$-th level of the tree, $\{\tau_{j,k}\}_{0 \leq k \leq n/2^j}$, conditioned on the higher levels of the tree, $\{\tau_{j',k}\}_{j'>j, 0 \leq k \leq n/2^{j'}}$, is the output distribution of mechanism described in Lem. 5.1, where the probabilities are

$$p_{j,k} := \Pr_{\tau \sim \Theta(D)} \left[ k \cdot 2^j + 1 \leq i^* \leq (k+1) \cdot 2^j \Big| \{\tau_{j',k}\}_{j'>j, 0 \leq k < n/2^{j'}} = \{\theta_{j',k}\}_{j'>j, 0 \leq k < n/2^{j'}} \right].$$

We now show a high probability bound on each of these probabilities. We have:

$$\Pr_{\tau \sim \Theta(D)} \left[ k \cdot 2^j + 1 \leq i^* \leq (k+1) \cdot 2^j \Big| \{\tau_{j',k}\}_{j'>j, 0 \leq k < n/2^{j'}} = \{\theta_{j',k}\}_{j'>j, 0 \leq k < n/2^{j'}} \right]$$

$$= \frac{\sum_{i=k \cdot 2^j + 1}^{(k+1) \cdot 2^j} \mathbf{Pr}_{\tau \sim \Theta(D)} \left[ \{\tau_{j',k}\}_{j'>j, 0 \leq k < n/2^{j'}} = \{\theta_{j',k}\}_{j'>j, 0 \leq k < n/2^{j'}} \Big| i^* = i \right]}{\sum_{i=1}^{n} \mathbf{Pr}_{\tau \sim \Theta(D)} \left[ \{\tau_{j',k}\}_{j'>j, 0 \leq k < n/2^{j'}} = \{\theta_{j',k}\}_{j'>j, 0 \leq k < n/2^{j'}} \Big| i^* = i \right]}$$

$$= \frac{\sum_{i=k \cdot 2^j + 1}^{(k+1) \cdot 2^j} \prod_{j'>j, 0 \leq k < n/2^{j'}} \exp\left( -\frac{\left(\tau_{j',k} - \mathbb{1}(k \cdot 2^{j'} + 1 \leq i^* \leq (k+1) \cdot 2^{j'})\right)^2}{2\sigma^2} \right)}{\sum_{i=1}^{n} \prod_{j'>j, 0 \leq k < n/2^{j'}} \exp\left( -\frac{\left(\tau_{j',k} - \mathbb{1}(k \cdot 2^{j'} + 1 \leq i^* \leq (k+1) \cdot 2^{j'})\right)^2}{2\sigma^2} \right)}$$

$$= \frac{\sum_{i=k \cdot 2^j + 1}^{(k+1) \cdot 2^j} \prod_{j,k: k \cdot 2^{j'} + 1 \leq i \leq (k+1) \cdot 2^{j'}} \exp\left( -\frac{\tau_{j',k}}{\sigma^2} \right)}{\sum_{i=1}^{n} \prod_{j,k: k \cdot 2^{j'} + 1 \leq i \leq (k+1) \cdot 2^{j'}} \exp\left( -\frac{\tau_{j',k}}{\sigma^2} \right)}$$

$$\leq \frac{2^j}{n} \cdot \frac{\max_{i \in [n]} \prod_{j,k: k \cdot 2^{j'} + 1 \leq i \leq (k+1) \cdot 2^{j'}} \exp\left( -\frac{\tau_{j',k}}{\sigma^2} \right)}{\min_{i \in [n]} \prod_{j,k: k \cdot 2^{j'} + 1 \leq i \leq (k+1) \cdot 2^{j'}} \exp\left( -\frac{\tau_{j',k}}{\sigma^2} \right)}$$

$$\leq \frac{2^j}{n} \cdot \exp\left( \frac{(\log n - j) \max_{j',k} \tau_{j',k} - \min_{j',k} \tau_{j',k}}{\sigma^2} \right).$$

With probability $1 - \delta/2$, by a union bound for all $2n$ pairs $j, k$ we have $|\tau_{j,k}| \leq \sqrt{2\ln(4n/\delta)}\sigma$, so the above bound is at most:

$$\frac{2^j}{n} \cdot \exp\left(\frac{(\log n - j)2\sqrt{2\ln(4n/\delta)}}{\sigma}\right).$$

If $\sigma \geq \frac{4\sqrt{2\ln(4n/\delta)}}{\ln 2}$ in turn this is at most:

$$\frac{2^j}{n} \cdot \sqrt{2}^{\log n - j} = \sqrt{\frac{2^j}{n}}.$$

Now, by Theorem C.2 and Observation 3.2 it suffices to show that conditioned on this probability $1 - \delta/2$ event, the binary tree mechanism satisfies $\left(O\left(\frac{\sqrt{\log(1/\delta)\log\log(1/\delta)}}{\sigma}\right), \delta/2\right)$-DP. For $\log n - 16e\log\log(16n/\delta) \leq j \leq \log n$, releasing $\tau_{j,k}$ satisfies $\left(O\left(\frac{\sqrt{\log(1/\delta)\log\log(n/\delta)}}{\sigma}\right), \delta/4\right)$-DP by the analysis of the (unamplified) Gaussian mechanism. For levels $0 \leq j < \log n - 16e\log\log(16n/\delta)$, our upper bound on the conditional probabilities and Lem. 5.1 with $\delta_0 = \delta/8n'\log n$ shows that, conditioned on the high-probability event, the distribution of the privacy loss of outputting $\{\tau_{j,k}\}_k$ conditioned on levels $j' > j$ satisfies $\left(O\left(\frac{\sqrt{\ln(n'/\delta)\ln(1/\delta)}}{\sigma\sqrt{n'}}\right), \delta/4\log n\right)$-DP, with $n' = \lceil\sqrt{\frac{n}{2^j}}\rceil$. By basic composition, the overall privacy loss distribution conditioned on the $1 - \delta/2$ probability event satisfies:

$$\left(O\left(\frac{\sqrt{\log(1/\delta)\log\log(1/\delta)}}{\sigma}\right) + O\left(\sum_{j=\log n - 16e\log\log(16n/\delta)}^{0} + \frac{2^{j/4}\ln(1/\delta)}{\sigma n^{1/4}}\right), \delta/2\right)\text{-DP}.$$

Here we use the upper bound on $\delta$ which is equivalent to $\log(n/\delta) = O(\log(1/\delta))$. We conclude by bounding the sum as:

$$\sum_{j=\log n - 16e\log\log(16n/\delta)}^{0} + \frac{2^{j/4}\ln(1/\delta)}{\sigma n^{1/4}}$$

$$\leq \sum_{l=0}^{\log n - 16e\log\log(16n/\delta)} + \frac{\ln(1/\delta)}{\sigma 2^{l/4}\sqrt{\ln(1/\delta)}} = \frac{\sqrt{\ln(1/\delta)}}{\sigma}\sum_{l=0}^{\log n - 16e\log\log(16n/\delta)} \frac{1}{2^{l/4}} = \frac{\sqrt{\ln(1/\delta)}}{\sigma(1 - 2^{-1/4})}.$$

$\square$

We now discuss how to extend the proof to a more general case. In other words, we choose some $\mathbf{y}$ with each row having 2-norm at most 1 for $D$, and then set $\mathbf{y}'$ for $D'$ to be $\mathbf{y}$ with the first row zeroed out. Then, $\mathbf{x}$ is chosen by shuffling the rows of $\mathbf{y}$ or $\mathbf{y}'$.

**Lemma C.4.** *Under the above setup, for some $k$ that divides $n$, consider the mechanism that chooses a random size $k$ equipartition of $[n]$, $P = (S_1, S_2, \ldots S_k)$ of $[n]$ according to some distribution and outputs $(\theta_1, \theta_2, \ldots, \theta_k)$, $\theta_i \sim N(\sum_{j \in S_i} \mathbf{y}_i, \sigma^2)$. Suppose for any two equipartitions $P, P'$, the probability of choosing $P$ is at most $c$ times the probability of choosing $P'$, and let $n' = \lfloor k/c \rfloor$.*

*Then, for any* $\delta \geq 2e^{-\frac{n'}{16e}}, \delta_0 \geq 0,$ *if* $\sigma = \Omega(\sqrt{\ln(1/\delta_0)})$ *then this mechanism satisfies* $\left(O\left(\frac{\sqrt{\ln(1/\delta_0)\ln(1/\delta)}}{\sigma\sqrt{n'}}\right), \delta + n'\delta_0\right)$*-DP.*

*Proof.* Recall that $\mathbf{y}_1$ is the example differing between $D$ and $D'$. By post-processing and quasi-convexity, we can instead analyze the mechanism that for each $S_i$, also publishes all but one element in $S_i$, and specifically for the $S_i$ including 1 (the sensitive element), the element of $S_i$ not published must be 1. This is equivalent to saying: without loss of generality we can assume $n = k$.

Next, the assumption on the distribution over $P$ implies that the distribution is in the convex hull of distributions over $P$ that deterministically choose $k - n'$ elements of $P$, with 1 being one of these $n'$ unchosen elements, and then uniformly shuffle the remaining $n'$ elements. In terms of privacy guarantees, each individual mechanism using one of these distributions is equivalent to $n'$ Gaussian mechanisms on shuffled elements. Then, by quasi-convexity, the privacy guarantees of this mechanism are no worse than those of a Gaussian mechanism over $n'$ shuffled elements. We conclude using the analysis of amplification by shuffling in (Feldman et al., 2022). □

Next, we use the following black-box reduction from $(\varepsilon, \delta)$-DP guarantees to high-probability privacy loss bounds:

**Lemma C.5** ((Kasiviswanathan & Smith, 2014)). *If a mechanism satisfies $(\varepsilon, \delta)$-DP, then the probability the privacy loss of its output exceeds $2\varepsilon$ is at most $\frac{\delta}{\varepsilon e^\varepsilon}$.*

Now, the high-level idea is that the $(\varepsilon, \delta)$-DP guarantee on outputting levels $j' > j$ implies a high-probability bound on the privacy loss of outputting these levels via Lem. C.5, which in turn implies a bound on $c$ in Lem. C.4 if we use the posterior distribution over shuffles as the distribution in that lemma. Then, we can use Lem. C.4 to get an $(\varepsilon, \delta)$-DP guarantee for round $j$ conditioned on the previous rounds, and as before the resulting $\varepsilon$ per level decays geometrically and we can use basic composition.

*Proof of Theorem 5.2.* By the upper bound on $\delta$, $\log(\text{poly}(n)/\delta) = O(\log(1/\delta))$. So, for any constant $c_1$ and another constant $c_2$ depending on $c_1$, releasing levels $\log n - c_1 \log \log 1/\delta$ to $\log n$ satisfies $\left(\frac{c_2\sqrt{\log(1/\delta)\log\log(1/\delta)}}{\sigma}, \delta/n^2\right)$-DP by analysis of the Gaussian mechanism.

Now, we will show by induction that releasing levels $j$ to $\log n$, $j \leq \log n - c_1 \log \log 1/\delta$, satisfies $(\varepsilon_j, \delta_j)$-DP for:

$$\varepsilon_j = \frac{c_2\sqrt{\log(1/\delta)\log\log(1/\delta)}}{\sigma} + \sum_{j'=\log n - c_1 \log \log(1/\delta)}^{j} \frac{c_3 \log(1/\delta)}{\sigma\sqrt{2^{\log n - j}}},$$

$$\delta_j = \frac{\delta}{n^2} \cdot \sum_{j'=\log n - c_1 \log \log(1/\delta)}^{j} (1 + 1/e)^{\log n - c_1 \log \log(1/\delta) - j}.$$

In the inequality on $\varepsilon_j$ we assume $c_1$ is sufficiently large. The base case of $j = \log n - c_1 \log \log 1/\delta$ holds by the aforementioned analysis of the Gaussian mechanism. Now, assuming releasing levels $j + 1$ to $\log n$ satisfies $(\varepsilon_j, \delta_j)$-DP, we will prove releasing levels $j$ to $\log n$ satisfies $(\varepsilon_j, \delta_j)$-DP. Consider the output distribution of level $j$, conditioned on the event that the privacy loss of releasing levels $j + 1$ to $\log n$ is at most 1. The privacy loss being at most 1 implies that conditioned on levels $j + 1$ to $\log n$'s output, no shuffle is more than $e$ times as likely as any other shuffle, and thus the same is true for equipartitions of the data into

the sums in level $j$. Then by Lem. C.4, level $j$ satisfies, say, $(\frac{c_3\sqrt{\log^2(1/\delta)}}{\sigma\sqrt{2^{\log n - j}}}, \delta/n^2)$-DP for some sufficiently large constant $c_3$, assuming $c_1$ is sufficiently large and $\sigma \geq c_4\sqrt{\log(1/\delta)\log\log(1/\delta)}$ for a sufficiently large constant $c_4$. We have

$$\varepsilon_{j+1} \leq \frac{c_2\sqrt{\log(1/\delta)\log\log(1/\delta)} + 4c_3\sqrt{\log(1/\delta)}}{\sigma}$$

So $\varepsilon_j < 1/2$ for all $j$, again assuming $\sigma \geq c_4\sqrt{\log(1/\delta)\log\log(1/\delta)}$ for sufficiently large $c_4$, by Lem. C.5 this event happens with probability at least $1 - \delta_{j+1}/e$. Then assuming releasing levels $j+1$ to $\log n$ satisfies $(\varepsilon_{j+1}, \delta_{j+1})$-DP by Thm. 4.3 and basic composition, we have proven releasing levels $j$ to $\log n$ satisfies $(\varepsilon_j, \delta_j)$-DP for

$$\varepsilon_j = \varepsilon_{j+1} + \frac{c_3\log(1/\delta)}{\sigma\sqrt{2^{\log n - j}}}, \delta_j = \delta_{j+1}(1 + 1/e) + \delta/n^2.$$

The claimed non-recursively defined values for $\varepsilon_j, \delta_j$ follow by unrolling the above recursive formula and plugging in the base case $j = \log n - c_1\log\log 1/\delta$. Now, the full binary tree mechanism with shuffling satisfies $(\varepsilon_0, \delta_0)$-DP for $\varepsilon_0 = O\left(\frac{\sqrt{\log(1/\delta)\log\log(1/\delta)}}{\sigma}\right), \delta_0 \leq \delta$ as desired. (Note that the between the constants $c_1, c_2, c_3, c_4$ there are no circular dependencies, i.e. there does exist a set of constants satisfying the assumptions in the proof.) □

## D  EXTENDING MMCC TO "$b$-MIN-SEP SAMPLING"

(Choquette-Choo et al., 2023a) analyzed the $b$-banded matrix mechanism under the following scheme, which we'll call "$b$-min-sep sampling": We partition the dataset $D$ into $b$ equal-size subsets, $D_1, D_2, \ldots D_b$. To compute $\mathbf{x}_i$, we use independently include each element of $D_{i \pmod b}$ (where we say $i \pmod b = b$ if $b$ divides $i$) with probability $bp$; here, we write the sampling probability in these rounds as $bp$ instead of $p$ to reflect the fact that the average example still participates in fraction $p$ of rounds in expectation for any choice of $b$.

We give a generalization of MMCC that analyzes the matrix mechanism under $b$-min-sep sampling, that matches the analysis of (Choquette-Choo et al., 2023a) when $\mathbf{C}$ is $b$-banded but can generalize to arbitrary lower triangular matrices. In other words, this generalization of MMCC subsumes the analysis in (Choquette-Choo et al., 2023a).

Note that if we want to analyze the privacy guarantee for an example in $D_i, i-1$, this is the same as analyzing the privacy guarantee for an example in $D_1$, if we use $\mathbf{C}$ with the first $i - 1$ rows/columns cut off. Then, without loss of generality we only need to state a privacy analysis for examples in $D_1$ - to get a privacy guarantee that holds for all examples simultaneously, for each $D_i$ we can compute a privacy guarantee using the above reduction, and then take the worst of these. Further, for some classes of matrices, such as Toeplitz matrices, the examples in $D_1$ will have the worst privacy guarantee and thus it suffices to only analyze these examples.

We now show Generalized-MMCC, given in Fig. 6, computes a valid privacy guarantee under $b$-min-sep sampling.

**Theorem D.1.** *Let $\varepsilon$ be the output of* Generalized-MMCC. *Then the matrix mechanism with matrix $\mathbf{C}$, $b$-min-sep sampling, sampling probability $p$, noise level $\sigma$ satisfies $(\varepsilon, \delta_1 + \delta_2)$-DP (for examples in $D_1$).*

---

**Algorithm 3** `Generalized-MMCC`

---

1: **Input:** Matrix $\mathbf{C}$, sampling probability $p$, noise standard deviation $\sigma$, probabilities $\delta_1, \delta_2$, min-sep $b$.
2: Delete all columns of $\mathbf{C}$ except columns $1, b+1, 2b+1 \ldots$
3: $\{\widetilde{p}_{i,j}\}_{i \in [n], j \in [\lceil n/b \rceil]} \leftarrow \text{GeneralizedProbabilityTailBounds}(\mathbf{C}, bp, \sigma, b\delta_1)$.
   $\triangleright$ $\widetilde{p}_{i,j}$ is a high-probability upper bound on the probability that an example participated in round $j$, conditioned on output in rounds $1$ to $i-1$.
4: $\widehat{p}_{i,j}^{(b)} = \widetilde{p}_{(i-1)b+1, (j-1)b+1}$
5: $\mathbf{C}_{i,j}^{(b)} = \left\| \mathbf{C}_{(i-1)b+1:ib, (j-1)b+1} \right\|_2$
6: **for** $i \in [\lceil n/b \rceil]$ **do**
7: $\quad PLD_i \leftarrow \text{PLD of } \mathcal{M}_{PMoG}(\{\widehat{p}_{i,j}^{(b)}\}_{j \in [\lceil n/b \rceil]}, \{\mathbf{C}_{i,j}^{(b)}\}_{j \in [\lceil n/b \rceil]})$.
8: $PLD \leftarrow \text{convolution of } \{PLD_i\}_{i \in [\lceil n/b \rceil]}$.
9: **return** $\min (\{\varepsilon : PLD \text{ satisfies } (\varepsilon, \delta_2)\text{-DP}\})$.

---

Figure 6: Extension of `MMCC` to $b$-min-sep sampling.

---

**Algorithm 4** `GeneralizedProbabilityTailBounds`$(\mathbf{C}, p, \sigma, \delta_1)$

---

1: **Input:** Matrix $\mathbf{C} \in \mathbb{R}^{m \times n}$, sampling probability $p$, noise standard deviation $\sigma$, probability $\delta_1$.
2: $\delta' = \frac{\delta_1}{2 \cdot (nnz(\mathbf{C}) - n)}$ $\quad\quad\quad\quad\quad\quad\quad\quad\quad \triangleright nnz$ is the number of non-zeros.
3: $z = \Phi^{-1}(1 - \delta')$ $\quad\quad\quad \triangleright$ Tail bound on normal distribution; here, $\Phi$ is the standard normal CDF.
4: **for** $i \in [m], j \in [n]$ **do**
5: $\quad$ **if** $\mathbf{C}_{i,j} = 0$ **then**
6: $\quad\quad \widetilde{p}_{i,j} = 1$
7: $\quad$ **else**
8: $\quad\quad s_{i,j} = \text{minimum } s \text{ s.t. } \mathbf{Pr}[\sum_{j' \leq i} x_{j'} \langle \mathbf{C}_{1:i-1,j}, \mathbf{C}_{1:i-1,j'} \rangle > s] \leq \delta', x_{j'} \overset{i.i.d.}{\sim} Bern(p)$
   $\triangleright s_{i,j}$ is a tail bound on the dot product of first $i-1$ entries of $\mathbf{Cx}$ and $\mathbf{C}_{1:i-1,j}$.
9: $\quad\quad \varepsilon_{i,j} = \frac{z\|\mathbf{C}_{1:i-1,j}\|_2}{\sigma} + \frac{2s_{i,j} - \|\mathbf{C}_{1:i-1,j}\|_2^2}{2\sigma^2}$
   $\triangleright \varepsilon_{i,j}$ is a tail bound on the privacy loss of a participation in round $j$ after outputting first $i-1$ rounds
10: $\quad\quad \widetilde{p}_{i,j} = \frac{p \cdot \exp(\varepsilon_{i,j})}{p \cdot \exp(\varepsilon_{i,j}) + (1-p)}$
11: **return** $\{\widetilde{p}_{i,j}\}_{i \in [m], j \in [n]}$.

---

Figure 7: Generalization of `ProbabilityTailBounds`.

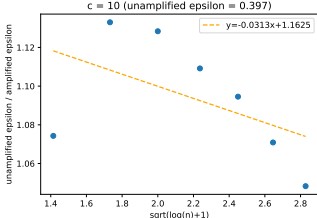 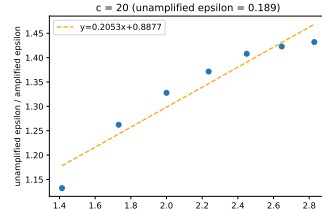 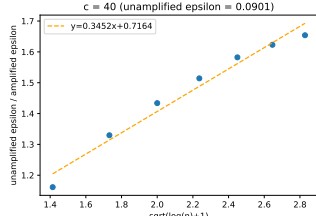

Figure 8: Plot of multiplicative improvement in $\varepsilon$ for the optimal continual counting matrix mechanism as a function of $\sqrt{\log(n)+1} \approx \|\mathbf{Ce}_1\|_2$. We plot $n = 2^i, i \in \{1, 2, \ldots, 7\}$. We use $\sigma = c\,\|\mathbf{Ce}_i\|_2$, so the $\varepsilon$ value in the unamplified single-participation setting is fixed. All $\varepsilon$ are for $\delta = 10^{-6}$.

*Proof.* The algorithm is almost the same as Thm. 4.3, so we just need to justify the key differences. In particular, we need to justify (1) the deletion of columns, (2) the choice of $\widetilde{p}_{i,j}^{(b)}$, and (3) the choice of $\mathbf{C}^{(b)}$.

(1) is justified by the proof of Theorem 4 in (Choquette-Choo et al., 2023a), which observes that the products of columns $j$ of $\mathbf{C}$ for which $j \pmod b \neq 1$ and the corresponding rows of $\mathbf{x}$ are independent of $D_1$, i.e. we can treat their products as public information. So it does not affect the privacy analysis to delete these rows/columns from $\mathbf{C}/\mathbf{x}$, and then view the resulting $\mathbf{x}$ as generated by i.i.d sampling every round with probability $bp$.

(2) and (3) are both justified if we use conditional composition over sequential mechanisms corresponding to $b$ rows of $\mathbf{Cx} + \mathbf{z}$ instead of a single row. Each of these sequential mechanisms is a VMoG mechanism, which Cor. B.6 allows us to reduce to the scalar PMoG mechanism defined in terms of $\mathbf{C}^{(b)}$ in `Generalized-MMCC`. The probabilities $\widetilde{p}^{(b)}$ are then valid to use in the conditional composition by the same argument as in Thm. 4.3, up to the adjustment to use $b\delta_1$ instead of $\delta_1$. This adjustment is valid, since we only use fraction $1/b$ of the values generated by `GeneralizedProbabilityTailBounds`, i.e. we are union bounding over $1/b$ as many "bad" events as in the original proof, so we can increase the allowed probability for each "bad" events by $b$ (which is implicitly done by increasing $\delta_1$ by $b$). □

One can verify that (i) for $b = 1$, `Generalized-MMCC` is equivalent to `MMCC`, and that (ii) if $\mathbf{C}$ is $b$-banded, `Generalized-MMCC` is equivalent to the privacy analysis in (Choquette-Choo et al., 2023a).

# E    MORE EMPIRICAL $\varepsilon$ COMPUTATIONS

(Fichtenberger et al., 2023; Henzinger et al., 2023) showed that a post-processing of the matrix mechanism using the following lower-triangular matrix achieves $1 + o(1)$ times the optimal $\ell_2^2$ error for prefix sums (without amplification): $\mathbf{C}_{i,j} = f(i - j)$, where $f$ is defined as

$$
f(k) = \begin{cases} 0, & \text{for } k < 0 \\ 1, & \text{for } k = 0 \\ f(k-1) \cdot \left(1 - \frac{1}{2k}\right), & \text{for } k > 0 \end{cases} .
$$

Similarly to the binary tree mechanism, we will consider the unamplified single-participation setting as a baseline. In this case, the sensitivity of this matrix mechanism is $\|\mathbf{Ce}_1\|_2$, i.e. the $\ell_2$-norm of the first column of $\mathbf{C}$. So again, setting $\sigma = c\,\|\mathbf{Ce}_1\|_2$ results in a fixed $\varepsilon$ for a fixed $\delta$. Our comparison will be applying the same matrix mechanism with subsampling probability $1/n$ and the same choice of $\sigma$.

In Fig. 8, we reproduce the plots in Fig. 2 but for this matrix mechanism instead of the binary tree mechanism. The $\ell_2$-norm of the columns of this matrix asymptotically are $\Theta(\sqrt{\log n})$; because of this, and to make a direct comparison to the binary tree mechanism easier, we use $\sqrt{\log(n)+1}$ as the x-axis and plot the least squares linear regression. Because the columns of this matrix are less orthogonal than those of the matrix for the binary tree mechanism, there is less benefit from amplification in this setting than the binary tree mechanism setting, so we use a larger range of values $c \in \{10, 20, 40\}$ for the noise multiplier to better demonstrate the behavior of the improvement in $\varepsilon$.

For sufficiently large $\sigma$, the improvement in $\varepsilon$ due to the amplification analysis is again roughly proportional to $\sqrt{\log(n)+1}$. For the same reasons as for the binary tree mechanism, the fit of the linear regression is better as $\sigma$ increases: here, because the columns of this matrix are less orthogonal on average, a larger value of $c$ is needed for the fit to improve. Here, the constant multiplier in the improvement is smaller; this makes sense as these matrices improve on the error of the binary tree mechanism by a constant, and thus the amount by which we can improve the privacy analysis of this matrix mechanism without violating lower bounds is smaller than for the binary tree mechanism.

## F  IMPLEMENTATION DETAILS

To implement the `MMCC` algorithm, we use the open-source Python library `dp_accounting.pld`[8]. We extend the class `dp_accounting.pld.pld_mechanism.AdditiveNoisePrivacyLoss` to create a class, `MixtureGaussianPrivacyLoss` that represents the privacy loss distribution of $\mathcal{M}_{MoG}$, which can be used along with other tools in the `dp_accounting.pld` library to implement `MMCC`. We discuss our implementation and some challenges here. The `dp_accounting.pld` library uses the convention that privacy losses are decreasing; we use the same convention in the discussions in this section for consistency. Our implementation is currently open-sourced as part of the `dp_accounting` library, and PLD accounting for MoG mechanisms can be done using `dp_accounting.pld.PLDAccountant` and `dp_accounting.dp_event.MixtureOfGaussiansDpEvent`.

### F.1  EXTENDING ADDITIVENOISEPRIVACYLOSS

In order to perform all the necessary computations in `MMCC`, we need to implement the following methods in `MixtureGaussianPrivacyLoss`:

1. A method to compute the CDF of the mixture of Gaussians distribution.
2. A method to compute the privacy loss at $x$.
3. An inverse privacy loss method, i.e. a method which takes $\varepsilon$ and computes the smallest $x$ achieving this $\varepsilon$.

Given the probabilities and sensitivities $\{p_1, p_2, \ldots, p_k\}$ and $\{c_1, c_2, \ldots, c_k\}$, as well as $\sigma$, the first two can easily be done by just summing the PDFs/CDFs of the Gaussians in the mixture. This takes at most $O(k)$ times the runtime of the corresponding method for the (subsampled) Gaussian mechanism.

The third is more problematic. For the subsampled Gaussian mechanism with sampling probability $p$ and sensitivity 1, the privacy loss function (under the remove adjacency) is:

$$\ln\left( p \exp\left( \frac{-2x-1}{2\sigma^2} \right) + 1 - p \right).$$

---

[8]https://github.com/google/differential-privacy/tree/main/python/dp_accounting/dp_accounting

This function is easily invertible. However, if we consider $\mathcal{M}_{MoG}(\{p, 1-p\}, \{c_1, c_2\})$, the privacy loss at $x$ is:

$$\ln\left(p\exp\left(\frac{-2c_1x - c_1^2}{2\sigma^2}\right) + (1-p)\exp\left(\frac{-2c_2x - c_2^2}{2\sigma^2}\right)\right).$$

Because this function includes the sum of two exponential functions of $x$, it is not easy to invert. We instead use binary search to get the smallest multiple of $\Delta_1$ which achieves the desired privacy loss, where $\Delta_1$ is a parameter we choose that trades off between efficiency and accuracy. That is, if $L$ is the privacy loss function, and we want to compute the inverse privacy loss of $y$, we return $x = \lceil L^{-1}(y)/\Delta_1 \rceil \cdot \Delta_1$. Note that by overestimating $x$, we also overestimate the privacy loss since we assume the privacy loss is decreasing. Hence this approximation is "pessimistic," i.e. does not cause us to report an $(\varepsilon, \delta)$-DP guarantee that is not actually satisfied by $\mathcal{M}_{MoG}$.

Note that using binary search requires a $O(\log(1/\Delta_1))$ multiplicative dependence on $\Delta_1$, that is not incurred for e.g. the subsampled Gaussian for which we can quickly compute the exact inverse privacy loss. Indeed, we observed that this inverse privacy loss method is the bottleneck for our implementation.

## F.2 EFFICIENTLY REPRESENTING PMoG AS MoG

As discussed in the previous section, the runtime of our implementation has a linear dependence on the number of components in the MoG. However, in MMCC, we are actually using PMoGs, which are MoGs with potentially $2^n$ components. So, even just listing the components can be prohibitively expensive.

We instead choose another approximation parameter $\Delta_2$, and round each entry of $\mathbf{C}$ up to the nearest multiple of $\Delta_2$. By Lemma B.4, this only worsens the privacy guarantee, i.e. any privacy guarantee we prove for the rounded version of $\mathbf{C}$ also applies to the original $\mathbf{C}$. After this rounding, the number of components in any MoG we compute the PLD of is at most $\lceil \max_i \left\| \mathbf{e}_i^\top \mathbf{C} \right\|_1 \rceil/\Delta_2 + n$ ($\max_i \left\| \mathbf{e}_i^\top \mathbf{C} \right\|_1$ is the maximum row norm of $\mathbf{C}$). Furthermore, we can compute the probabilities/sensitivities efficiently since we are working with PMoGs. In particular, for each $\widetilde{p}_{i,j}, \mathbf{C}_{i,j}$ pair, we can construct the probability mass function (PMF) of the random variable that is $\mathbf{C}_{i,j}$ w.p. $\widetilde{p}_{i,j}$ and 0 otherwise, and then take the convolution of all such PMFs for a row to get the PMF of the discretized sensitivity for the PMoG. For each row, this can be done in at most $n - 1$ convolutions, each convolution between two PMFs that have support size at most 2 and $\max_i\lceil \left\| \mathbf{e}_i^\top \mathbf{C} \right\|_1 /\Delta_2 \rceil + n$. So the convolutions can be done in time $O(\max_i\lceil \left\| \mathbf{e}_i^\top \mathbf{C} \right\|_1 /\Delta_2 \rceil + n)$, i.e. our overall runtime is $O(n^2 \max_i\lceil \left\| \mathbf{e}_i^\top \mathbf{C} \right\|_1 /\Delta_2 \rceil + n^3)$, i.e. polynomial instead of exponential in $n$ if e.g. all entries of $\mathbf{C}$ are bounded by a constant. By doing the convolutions in a divide-and-conquer fashion, and using FFT for the convolutions, we can further improve the runtime to $\widetilde{O}(n \max_i\lceil \left\| \mathbf{e}_i^\top \mathbf{C} \right\|_1 /\Delta_2 \rceil + n^2)$, i.e. nearly linear in the input size and $1/\Delta_2$ if the entries of $\mathbf{C}$ are bounded by a constant.

## F.3 COMPUTING $s_{i,j}$

Similar to computing the probabilities and sensitivities for the PMoGs, any overestimate of $s_{i,j}$ can be used in place of $s_{i,j}$ to get a valid privacy guarantee from MMCC by Lemma B.3. Since $s_{i,j}$ only appears in a lower order term in the definition of $\varepsilon_{i,j}$, a weaker tail bound will not affect the privacy guarantee as much. So, in our implementation, we use the following simple and efficient approximation: We use the binomial CDF to obtain an exact tail bound $t$ on $\|\mathbf{x}_{1:i}\|_1 = \sum_{j' \leq i} x_{j'}$ in the definition of $s_{i,j}$. We then take the sum of the $t$ largest values of $\langle \mathbf{C}_{1:i-1,j}, \mathbf{C}_{1:i-1,j'} \rangle$ to be our overestimate of $s_{i,j}$.

### F.4 COMPUTING ALL ROW PLDS

Putting this all together, we must compute $n$ PLDs in `MMCC`, one for each row of $\mathbf{C}$. Though only an $O(n)$ overhead in runtime over computing a single PLD, this $O(n)$ overhead is undesirable as each PLD computation is already quite expensive due to the aforementioned difficulties. However, this component is embarrassingly parallel, which we leverage to massively speed up runtimes.

Note that for some special classes of matrices, we will have that multiple rows share the same PLD, which also allows us to dramatically speed up the calculation even without parallelization. For example, this is the case for the binary tree mechanism due to symmetry, as well for as $b$-banded Toeplitz $\mathbf{C}$ due to the fact that rows $2b - 1$ to $n$ of $\widetilde{p}$ and $\mathbf{C}$ are the same (up to an offset in indices that doesn't affect the PLD).

### F.5 APPLICATIONS BEYOND MATRIX MECHANISMS

We believe that MoG mechanisms/`MixtureGaussianPrivacyLoss` are useful analytic tools for privacy analysis of mechanisms beyond the matrix mechanism. We discuss two examples here.

**Privacy amplification via iteration on linear losses:** Consider running DP-SGD with sampled minibatches. To get a $(\varepsilon, \delta)$-DP guarantee, we can compute the PLD for the subsampled Gaussian mechanism, and then compose this PLD with itself $n$ times. For general non-convex losses, this accounting scheme is tight, even if we only release the last iterate.

For linear losses, we can give a better privacy guarantee for releasing only the last iterate, similarly to (Feldman et al., 2018): Releasing the last iterate is equivalent in terms of privacy guarantees to a Gaussian mechanism with random sensitivity $Binom(n, p)$ and variance $n\sigma^2$. Using `MixtureGaussianPrivacyLoss` we can get tight $(\varepsilon, \delta)$-DP guarantees for this mechanism. Empirically, we found that these can be a lot tighter than composition of subsampled Gaussians. For example, using $n = 128, p = 1/128, \sigma = 1$ we found that composition of subsampled Gaussians gives a proof of $(.806, 10^{-6})$-DP, whereas analyzing the last iterate as a MoG mechanism gives a proof of $(.291, 10^{-6})$-DP. We conjecture a similar improvement is possible for all convex losses, rather than linear losses.

**Tight group privacy guarantees for DP-SGD:** Consider analyzing the privacy guarantees of DP-SGD under group privacy. That is, we want to give a privacy guarantee for pairs of databases differing in $k > 1$ examples. One way of doing this is to compute a DP guarantee for $k = 1$, then use an example-to-group privacy theorem such as that of (Vadhan, 2017), which shows an $(\varepsilon, \delta)$-DP mechanism satisfies $(k\varepsilon, k\exp(k\varepsilon)\delta)$-DP for groups of size $k$. This is overly pessimistic, since the black-box theorem doesn't account for the specific structure of the mechanism. We can instead get relatively tight guarantees via `MixtureGaussianPrivacyLoss`: If each example is sampled independently, then the privacy loss of a group of $k$ examples in each round of DP-SGD is dominated by a Gaussian mechanism with sensitivity $Binom(k, p)$. Then, we can use `MixtureGaussianPrivacyLoss` to analyze the composition of $n$ of these MoG mechanisms. Further, note that e.g. in the case where we instead sample a random batch of size $B$ in each round (i.e. different examples' participations within the same round are no longer independent), we can still use `MixtureGaussianPrivacyLoss` to get a tight analysis by adjusting the sensitivity random variable used. See the follow-up note (Ganesh, 2024) for more details.

