# PRIVACY AMPLIFICATION FOR MATRIX MECHANISMS

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

 4: Comparison of TMSE for prefix sums from independent noise (the noise + analysis used in DP-SGD), and Toeplitz matrices analyzed via `MMCC` and the analysis from Choquette-Choo et al. (2023a)

### 6.3 CORRELATED NOISE AND AMPLIFICATION BEATS INDEPENDENT NOISE FOR SMALL $\varepsilon$

We show that `MMCC` enables correlated-noise mechanisms with better squared error than independent noise at much smaller $\varepsilon$ than past work. We consider the problem of computing prefix sums for $n = 1024$ rounds, with sampling probability $p = 1/32$ in each round, i.e. each example participates 32 times in expectation. In Fig. 4, we compare the $\ell_2^2$-error (i.e. total mean squared error/TMSE) of the estimate of all prefix sums of $\mathbf{x}$ as a function of $\varepsilon$ for (i) independent noise analyzed with standard PLD accounting (i.e., DP-SGD-style noise + analysis), (ii) a Toeplitz matrix mechanism analyzed with `MMCC`, where $\mathbf{C}$ is a simple matrix whose first diagonal is some "decay constant" $a$, second diagonal is $\sqrt{1 - a^2}$, and $\mathbf{C}$ is zero everywhere else, and (iii) the same Toeplitz matrices but analyzed using results in Choquette-Choo et al. (2023a) instead. Prior work was unable to show in this setting that any correlated noise matrix mechanism achieves better TMSE than independent noise mechanisms for $\varepsilon \leq 0.5$ (see Table 3 of Choquette-Choo et al. (2023a)). In contrast, in Fig. 4 we see that even with a somewhat arbitrary choice of $\mathbf{C}$, we are able to achieve noticeably smaller TMSE than independent noise for $\varepsilon$ as small as $\varepsilon \approx 0.03$.

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

 such that $\|\mathbf{c}_i\|_2 = \|\mathbf{c}'_i\|_2$ for all $i$ and $\langle \mathbf{c}_i, \mathbf{c}_j \rangle \leq \langle \mathbf{c}'_i, \mathbf{c}'_j \rangle$ for all $i, j$ pairs. Then the PLD of $\mathcal{M}_{VMoG}(\{p_1, p_2, \ldots, p_k\}, \{\mathbf{c}_1, \mathbf{c}_2, \ldots, \mathbf{c}_k\})$ is dominated by the PLD of $\mathcal{M}_{VMoG}(\{p_1, p_2, \ldots, p_k\}, \{\mathbf{c}'_1, \mathbf{c}'_2, \ldots, \mathbf{c}'_k\})$.*

*Proof.* In the remove adjacency case, letting $\mathbf{s}$ be the random sensitivity for $D$, the $\varepsilon$-hockey stick divergence between the output distributions of

$$\mathcal{M}_{VMoG}(\{p_1, p_2, \ldots, p_k\}, \{\mathbf{c}_1, \mathbf{c}_2, \ldots, \mathbf{c}_k\})$$

is:

$$\mathbb{E}_{\mathbf{s}, \mathbf{z} \sim N(\mathbf{s}, \sigma^2 \cdot \mathbb{I})} \left[ \max \left\{ 1 - \frac{e^\varepsilon}{\sum_i p_i \exp \left( \frac{2\langle \mathbf{z}, \mathbf{c}_i \rangle - \|\mathbf{c}_i\|_2^2}{2\sigma^2} \right)}, 0 \right\} \right]$$

$$= \sum_i p_i \mathbb{E}_{\mathbf{z} \sim N(\mathbf{0}, \sigma^2 \cdot \mathbb{I})} \left[ \max \left\{ 1 - \frac{e^\varepsilon}{\sum_j p_j \exp \left( \frac{2\langle \mathbf{z}, \mathbf{c}_j \rangle + 2\langle \mathbf{c}_i, \mathbf{c}_j \rangle - \|\mathbf{c}_j\|_2^2}{2\sigma^2} \right)}, 0 \right\} \right]$$

$$\leq \sum_i p_i \mathbb{E}_{\mathbf{z} \sim N(\mathbf{0}, \sigma^2 \cdot \mathbb{I})} \left[ \max \left\{ 1 - \frac{e^\varepsilon}{\sum_j p_j \exp \left( \frac{2\langle \mathbf{z}, \mathbf{c}'_j \rangle + 2\langle \mathbf{c}'_i, \mathbf{c}'_j \rangle - \|\mathbf{c}'_j\|_2^2}{2\sigma^2} \right)}, 0 \right\} \right].$$

Which is exactly the $\varepsilon$-hockey stick divergence between the output distributions of

$$\mathcal{M}_{VMoG}(\{p_1, p_2, \ldots, p_k\}, \{\mathbf{c}'_1, \mathbf{c}'_2, \ldots, \mathbf{c}'_k\}).$$

The add adjacency case follows by Lem. B.1. $\qquad\square$

As a corollary, we have a "vector-to-scalar reduction" for MoG mechanisms:

**Corollary B.3.** *The PLD of $\mathcal{M}_{VMoG}(\{p_1, p_2, \ldots, p_k\}, \{\mathbf{c}_1, \mathbf{c}_2, \ldots, \mathbf{c}_k\})$ is dominated by the PLD of $\mathcal{M}_{VMoG}(\{p_1, p_2, \ldots, p_k\}, \{\|\mathbf{c}_1\|_2, \|\mathbf{c}_2\|_2, \ldots, \|\mathbf{c}_k\|_2\})$.*

*Proof.* This follows from Lem. B.2 and the Cauchy-Schwarz inequality. $\qquad\square$

The following proof will allow us to simplify the distribution of the sensitivity random variable in a MoG mechanism:

**Lemma B.4.** *Let $\{p_1, p_2, \ldots p_k\}, \{c_1, c_2, \ldots, c_k\}$ and $\{p'_1, p'_2, \ldots p'_{k'}\}, \{c'_1, c'_2, \ldots c'_{k'}\}$ be such that for all $T$, $\sum_{i:c'_i \geq T} p'_i \geq \sum_{i:c_i \geq T} p_i$. In other words, the random variable induced by $\{p_i\}_i, \{c_i\}_i$ is stochastically dominated by the random variable induced by $\{p'_i\}_i, \{c'_i\}_i$. We also assume $c_i, c'_i \geq 0$ for all $i$.*

*Then the PLD of $\mathcal{M}_{MoG}(\{p_1, p_2, \ldots p_k\}, \{c_1, c_2, \ldots, c_k\})$ is dominated by the PLD of $\mathcal{M}_{MoG}(\{p'_1, p'_2, \ldots p'_{k'}\}, \{c'_1, c'_2, \ldots c'_{k'}\})$.*

*Proof.* By Lem. B.1 we only need to prove the lemma under the remove adjacency. The privacy loss of outputting $x$ for the first MoG mechanism is:

$$\ln \left( \sum_i p_i \exp \left( \frac{2xc_i - c_i^2}{2\sigma^2} \right) \right),$$

i.e. is increasing in $x$ (since all $c_i \geq 0$), and the same is true for the second MoG mechanism. This implies a MoG mechanism $\mathcal{M}$ satisfies $(\varepsilon, \delta)$-DP if and only if for all $t$:

$$\mathbf{Pr}_{x \sim \mathcal{M}(D)}[x \geq t] \leq e^\varepsilon \cdot \mathbf{Pr}_{x \sim \mathcal{M}(D')}[x \geq t] + \delta.$$

Then to show a MoG mechanism $\mathcal{M}$ is dominated by a MoG mechanism $\mathcal{M}'$, it suffices to show for all $t$:

$$\mathbf{Pr}_{x \sim \mathcal{M}(D)}[x \geq t] \leq \mathbf{Pr}_{x \sim \mathcal{M}(D')}[x \geq t].$$

Letting $s$ be the random sensitivity and $z$ be the Gaussian noise in the MoG mechanism for $\mathcal{M}$, we have:

$$\mathbf{Pr}_{x \sim \mathcal{M}(D)}[x \geq t] = \mathbf{Pr}_{s,z \sim \mathcal{M}(D)}[s + z \geq t] = \mathbb{E}_{z \sim N(0,\sigma^2)}[\mathbf{Pr}_s[s \geq t - z].$$

In turn, if $s$, the sensitivity for $\mathcal{M}$ is stochastically dominated by $s'$, the sensitivity for $\mathcal{M}'$, then for any $t, z$ we have $\mathbf{Pr}_s[s \geq t - z] \leq \mathbf{Pr}_{s'}[s' \geq t - z]$ and so

$$\mathbf{Pr}_{x \sim \mathcal{M}(D)}[x \geq t] \leq \mathbf{Pr}_{x \sim \mathcal{M}(D)}[x \geq t],$$

completing the proof. □

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

Let **x** be the (random) vector used in the matrix mechanism. In the analysis only, we can assume we also release which examples (except for the sensitive example) participate in all rounds; this can only worsen the DP guarantee of the mechanism. Then, the matrix mechanism is a VMoG mechanism, and by Lem. B.2, it suffices to consider the scalar case, i.e. for each $i$, independently

**Algorithm 2** `ProbabilityTailBounds`$(\mathbf{C}, p, \sigma, \delta_1)$

---

1: **Input:** Matrix $\mathbf{C}$, sampling probability $p$, noise standard deviation $\sigma$, probability $\delta_1$.
2: $\delta' = \frac{\delta_1}{2 \cdot (nnz(\mathbf{C}) - n)}$             $\triangleright$ $nnz$ is the number of non-zeros.
3: $z = \Phi^{-1}(1 - \delta')$     $\triangleright$ Tail bound on normal distribution; here, $\Phi$ is the standard normal CDF.
4: **for** $i, j \in [n]$ **do**
5:      **if** $\mathbf{C}_{i,j} = 0$ **then**
6:          $\widetilde{p}_{i,j} = 1$
7:      **else**
8:          $s_{i,j} = $ minimum $s$ s.t. $\mathbf{Pr}[\sum_{j' \leq i} x_{j'} \langle \mathbf{C}_{1:i-1,j}, \mathbf{C}_{1:i-1,j'} \rangle > s] \leq \delta', x_{j'} \overset{i.i.d.}{\sim} Bern(p)$
         $\triangleright$ $s_{i,j}$ is a tail bound on the dot product of first $i - 1$ entries of $\mathbf{Cx}$ and $\mathbf{C}_{1:i-1,j}$.
9:          $\varepsilon_{i,j} = \frac{z \|\mathbf{C}_{1:i-1,j}\|_2}{\sigma} + \frac{2s_{i,j} - \|\mathbf{C}_{1:i-1,j}\|_2^2}{2\sigma^2}$
         $\triangleright$ $\varepsilon_{i,j}$ is a tail bound on the privacy loss of a participation in round $j$ after outputting first $i - 1$
     rounds
10:          $\widetilde{p}_{i,j} = \frac{p \cdot \exp(\varepsilon_{i,j})}{p \cdot \exp(\varepsilon_{i,j}) + (1 - p)}$
11:      **end if**
12: **end for**
13: **return** $\{\widetilde{p}_{i,j}\}_{i,j \in [n]}$.

---

Figure 5: Algorithm for computing $\widetilde{p}_{i,j}$, tail bound on conditional probability of participating in round $j$ given first $i - 1$ outputs.

$\mathbf{x}_i = 1$ w.p. $p$ and $\mathbf{x}_i = 0$ otherwise. Under these assumptions, the value of $\mathbf{x}$ for $D'$ is deterministic (in the non-adaptive setting), i.e. we can without loss of generality assume for $D'$ that $\mathbf{x} = \mathbf{0}$. So, let $\Theta(D), \Theta(D')$ denote the distributions of $\mathbf{Cx} + \mathbf{z}, \mathbf{z}$ (i.e., the output of the matrix mechanisms run on $D$ and $D'$) respectively. We abuse notation and let $\mathbf{Pr}$ denote likelihood as well as a probability.

Our proof proceeds in three high-level steps:

1. We show the matrix mechanism is dominated by a sequence of adaptively chosen MoG mechanisms.
2. We show each of the adaptively chosen MoG mechanisms is further dominated by a PMoG mechanism.
3. We show these PMoG mechanisms are with high probability dominated by the PMoG mechanisms in MMCC, and then apply Theorem C.2.

**Step 1 (matrix mechanism dominated by sequence of MoG mechanisms):** By Observation 3.2, the matrix mechanism is the same as the composition over $i$ of outputting $\theta_i$ sampled from its distribution conditioned on $\theta_{1:i-1}$. Let $\theta_{1:i-1}$ denote the output of rounds 1 to $i-1$ of the matrix mechanism. Let $S_i$ denote $\{j \in [i] : \mathbf{x}_i = 1\}$. Abusing notation to let $\mathbf{Pr}$ denote a likelihood, the likelihood of $\mathcal{M}(D)$ outputting $\theta_i$ in round $i$ conditioned on $\theta_{1:i-1}$ is:

$$\sum_{T \subseteq [i]} \Pr_{\tau \sim \Theta(D)}[S_i = T | \tau_{1:i-1} = \theta_{1:i-1}] \Pr_{\tau_i \sim N(\sum_{j \in T} \mathbf{C}_{i,j} \mathbf{x}_j, \sigma^2 \cdot \mathbb{I})}[\tau_i = \theta_i]$$

The likelihood of $\mathcal{M}(D')$ outputting $\theta_i$ in round $i$ (conditioned on $\theta_{1:i-1}$, which doesn't affect the likelihood since since each coordinate of $\theta$ is independent when sampled from $\mathcal{M}(D')$) is

$$\Pr_{\tau_i \sim N(\mathbf{0}, \sigma^2 \cdot \mathbb{I})}[\tau_i = \theta_i].$$

In other words, the distribution of $\theta_i$ conditioned on under $\mathcal{M}(D), \mathcal{M}(D')$ is exactly the same as the pairs of output distributions given by the MoG mechanism

$$\mathcal{M}_{MoG}\left(\left\{\Pr_{\tau \sim \Theta(D)}[S_i = T | \tau_{1:i-1} = \theta_{1:i-1}]\right\}_{T \subseteq [i]}, \{\sum_{j \in T} \mathbf{C}_{i,j}\}_{T \subseteq [i]}\right).$$

Then by Lem. C.1 (composition preserves domination), the PLD of the matrix mechanism is dominated by the PLD of the composition of the (adaptively chosen) MoG mechanisms

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

(0, \left\|\mathbf{C}_{1:i-1,j}\right\|_2^2\sigma^2)$ and thus tail bounded by $z\left\|\mathbf{C}_{1:i-1,j}\right\|_2\sigma$ with the same probability by definition. A union bound over these two events gives the desired tail bound on $\langle\theta_{1:i-1}, \mathbf{C}_{1:i-1,j}\rangle$. □

**From non-adaptive to adaptive:** The reduction from the adaptive to the non-adaptive setting follows from the following lemma:

**Lemma C.4.** *Fix some unit vector $\mathbf{u}$. Let $L$ be the worst-case privacy loss random variable for a $n$-round matrix mechanism where for $D$, each row of $\mathbf{x}$ is $\mathbf{u}$ with probability $p$ and $\mathbf{0}$ with probability $1 - p$ (and for $D'$, $\mathbf{x} = \mathbf{0}$). Let $L'$ be the privacy loss random variable for an $n$-round matrix mechanism where the first $n - 1$ rows of $\mathbf{x}$ are chosen to be $\mathbf{u}$ with probability $p$ and $\mathbf{0}$ with probability $1 - p$, and the last row is an adaptively chosen unit vector $\mathbf{u}_n$ with probability $p$ and $\mathbf{0}$ with probability $1 - p$. Then $L$ dominates $L'$.*

*Proof.* The PLD of the first $n - 1$ rounds is the same for both mechanisms. From the proof of Thm. 4.3, if the output for the first $n - 1$ rounds is $\theta_{1:n-1}$, we know the PLD of the last round is the PLD of

$$\mathcal{M}_{VMoG}\left(\{\Pr[S_n = T|\theta_{1:n-1}]\}_{T\subseteq[n]}, \{\mathbf{u}\cdot\sum_{j\in T}\mathbf{C}_{n,j}\}_{T\subseteq[n]}\right)$$

for the first mechanism, and the PLD of

$$\mathcal{M}_{VMoG}\left(\{\mathbf{Pr}[S_n = T | \theta_{1:n-1}]\}_{T \subseteq [n]}, \{\mathbf{u} \cdot \sum_{j \in T \setminus [n]} \mathbf{C}_{n,j} + \mathbf{u}_n \cdot \mathbb{1}(n \in T)\}_{T \subseteq [n]}\right)$$

for the latter. In particular, note that for any output from the first $n - 1$ rounds, the probabilities are the same for both VMoG mechanisms. Then by Lem. B.2 and Lem. B.4 the first VMoG mechanism dominates the second. The lemma follows by Lem. C.1. $\qquad\square$

By induction, this implies the non-adaptive input sequence where each row of $\mathbf{x}$ is either $\mathbf{0}$ or $\mathbf{u}$ dominates the worst-case adaptive input sequence. This input sequence is the same as the scalar case we reduce to in the proof of Thm. 4.3 for non-adaptive mechanisms, so the privacy bound computed by MMCC for non-adaptive mechanisms holds for adaptive ones as well.

### C.3   PROOF OF LEM. 5.1

*Proof.* Since each $0 \le p_i \le 1/n'$, the mechanism is the same as the following: For each example we choose a subset $S \subseteq [n]$ of size $n'$ according to some distribution that is a function of the $p_i$, and then choose $i$ uniformly at random from the elements of $S$, and include the example in the $i$th subset. By quasi-convexity of approximate DP, it suffices to prove the DP guarantee for a fixed choice of $S$. For any fixed choice of $S$, the mechanism is equivalent to the shuffled Gaussian mechanism over $n'$ coordinates. Each unshuffled Gaussian mechanism satisfies $(\frac{\sqrt{2\ln(1.25/\delta_0)}}{\sigma}, \delta_0)$-DP, and then the lemma follows by the amplification via shuffling statement of Theorem 3.8 of Feldman et al. (2022). $\qquad\square$

### C.4   PROOF OF THEOREM 5.2

We first analyze a simplified case where $\mathbf{x}_i = 0$ if $i \ne i^*$, and otherwise $\mathbf{x}_i = 1$ for $D$ and $\mathbf{x}_i = 0$ for $D'$. We later give a proof for the general case.

*Proof of Theorem 5.2 in simplified case.* Let $\tau_{j,k}$ be the value of the noisy sum $\sum_{k \cdot 2^j + 1 \le i \le (k+1) \cdot 2^j} \mathbf{x}_i + \mathbf{z}_{j,k}$, $\tau = \{\tau_{j,k}\}_{0 \le j \le \log n, 0 \le k < n/2^j}$ and let $\Theta(D)$ be the distribution of these values under dataset $D$. We consider a single sensitive example; let $i^*$ be the (random) coordinate of $\mathbf{x}_{i^*}$ that this example contributes to.

Now, again abusing notation to let $\mathbf{Pr}$ denote a likelihood, we have for any $j$:

$$\underset{\tau \sim \Theta(D)}{\mathbf{Pr}}\left[\{\tau_{j,k}\}_{0 \le k < n/2^j} = \{\theta_{j',k}\}_{j' > j, 0 \le k < n/2^{j'}} \middle| \{\tau_{j',k}\}_{j' > j, 0 \le k < n/2^{j'}} = \{\theta_{j',k}\}_{j' > j, 0 \le k < n/2^{j'}}\right] \propto$$

$$\sum_{0 \le k^* \le n/2^j} \underset{\tau \sim \Theta(D)}{\mathbf{Pr}}\left[\{\tau_{j,k}\}_{0 \le k < n/2^j} = \{\theta_{j',k}\}_{j' > j, 0 \le k < n/2^{j'}} \middle| k^* \cdot 2^j + 1 \le i^* \le (k^* + 1) \cdot 2^j\right] \cdot$$

$$\underset{\tau \sim \Theta(D)}{\mathbf{Pr}}\left[k^* \cdot 2^j + 1 \le i^* \le (k^* + 1) \cdot 2^j \middle| \{\tau_{j',k}\}_{j' > j, 0 \le