# OpenReview forum: "Privacy Amplification for Matrix Mechanisms"
_ICLR.cc/2024/Conference — ICLR 2024 spotlight_

### Official Review · Reviewer_aP5d · 2023-10-29

**Soundness:** 3 good
**Presentation:** 3 good
**Contribution:** 3 good
**Rating:** 8
**Confidence:** 4

**Summary:**

Given a set of workload queries, the matrix mechanism identifies a set of basis vectors, adds noise to these basis vectors, and then computes DP answers to workload queries via linear combinations of the perturbed basis vectors. This unique algorithm structure poses a challenge to quantifying the privacy amplification by subsampling in matrix mechanism -- each subsampled sensitive data is possibly reused multiple times in computing different basis vectors (in which case the final noise added to each query is correlated). Due to the lack of such privacy amplification results in prior works, the DP-FTRL algorithm (which uses matrix mechanism as a crucial component) in general does not exhibit better performance than the DP-SGD algorithm.

- In this paper, the authors prove novel bounds for privacy amplification by subsampling for matrix mechanism. The proof decomposes the output distribution into the composition of multiple mixtures of Gaussian mechanisms, where each mixture component is the conditional distribution of the next basis vector given the participation histories of sensitive data and noisy observations in prior rounds.
- By using this analysis, the authors improve the existing unamplified privacy guarantee for the binary-tree-FTRL algorithm under shuffling by a multiplicative factor of $\Omega(\sqrt{\log(n)})$ where $n$ is the size of the dataset. This improvement is supported by numerical experiments.
- Finally, the authors show that this improved privacy analysis enables performance gain for the DP binary-tree-FTRL algorithm under small $\varepsilon$ under the CIFAR-10 dataset centralized setup.

**Strengths:**

- Novel bounds for privacy amplification by subsampling in matrix mechanisms. The proof involves several interesting new techniques, such as conditional composition and the analysis of the mixture of Gaussians mechanism. These techniques are of independent interest and may shed light on privacy amplification by subsampling/shuffling for other algorithms in the literature.

- Comparison with related works are discussed thoroughly, making it easy to follow the significance of the results.

**Weaknesses:**

- The computation cost for the privacy bounds seems high. Namely, the authors need to compute the privacy loss random variable for the product mixture of the Gaussians mechanism, which involves combinatorially many mixture components. Such procedures seem likely to suffer from high computation cost and numerical instability (especially under a small number of rounds or a small $\varepsilon$).

**Questions:**

- Could the authors discuss more about the computation cost and numerical stability of the proved privacy bound? See weakness for more details.

- Related to the above question, given an arbitrary pair of $\varepsilon, \delta$ values, how computationally feasible is it to compute the appropriate noise scale that ensures $(\varepsilon, \delta)$-DP of the proposed algorithm?

- In the proof of Lemma B.4, page 15, the third inequality from top $Pr_{x\sim M(D)}[x\geq t]\leq Pr_{x\sim M(D')}[x\geq t]$, is there a typo and $M(D')$ is meant to be $M'(D)$? I'd also like more clarifications regarding why this inequality suffices to show the MoG mechanism $\mathcal{M}$ is dominated by a MoG mechanism $\mathcal{M'}$. Specifically, how this inequality suffice to bound the term $-e^{\varepsilon}Pr_{x\sim M'(D')}[x\geq t]$ in the dominance term $H_{\varepsilon}(M'(D), M'(D'))$.

---

> ### Author Response · Authors · 2023-11-18
> **Response to reviewer aP5d**
>
> Thank you for the supportive and thoughtful review. Below we address the weaknesses/questions raised by the reviewer.
>
> **“Could the authors discuss more about the computation cost and numerical stability of the proved privacy bound?”**
>
> In terms of computation cost, in Appendix F.2, we address the challenge of handling the potentially $2^n$ mixture components mentioned in the reviewer’s weaknesses. Specifically, by rounding the entries of $C$ to the nearest multiple of some discretization parameter $\Delta$, we can substantially reduce the number of mixture components at a small cost in accuracy. While we have not proven the asymptotic runtime of MMCC, we believe with the above discretization it should be $O(n^3 + n^2 \log^2 n \cdot f)$, where $f$ is some function (independent of $n$) of the above discretization and approximation parameters used in the PLD accounting. Furthermore, many parts of our implementation can be vectorized or parallelized.
>
> In addition, for some specific classes of matrices, the computation can be dramatically sped up. For example, in Appendix F.4 we discuss how the calculation is much faster for the binary tree mechanism and for banded Toeplitz C.
>
> For numerical stability, in our implementation, the issues which were inherent to our approach (i.e. that wouldn’t appear when doing PLD accounting for DP-SGD) had to do with the fact that the mixtures of Gaussians arising from product distributions may have components with very small probabilities, even when we use the discretization trick in Appendix F.2 to reduce the number of components. In this case, we can always move these small probabilities around and absorb the cost in our delta term. That is, we can take components with at most, say, $10^{-20}$ of the probability mass in the mixture of Gaussians, delete these components, and increase the probability mass of a different component to make sure the total probability remains 1. Each time we do this, we pay a cost of $10^{-20}$ in the final $\delta$ we report, but this is ultimately negligible compared to e.g. the choice of $\delta = 10^{-6}$ we used throughout the paper.
>
> If there are specific questions the reviewer has about the runtime/implementation that were not discussed here, we would be happy to discuss them in followups.
>
> **“Related to the above question, how computationally feasible is it to compute the appropriate noise scale that ensures ($\epsilon, \delta$)-DP of the algorithm?”**
>
> The best approach we know of is to use binary search to (approximately) invert the function given by MMCC to compute $\epsilon$ given $\sigma$. So, in the aforementioned settings where MMCC can be run very quickly, one would need to pay a $\approx \log 1 / \Delta$ multiplicative factor to compute $\sigma$ up to precision $\Delta$, but the overall procedure is still quick. For general matrices, a single run of MMCC can take several hours or longer for $n \approx 2000$, and the $\log 1 / \Delta$ overhead may make this binary search infeasible.
>
> **Clarifications on Lemma B.4**
>
> Yes, this is a typo! Apologies for any confusion this caused and thanks for pointing this out; we will fix it in the revision.
>
> As for the clarification on why the stated inequality suffices to prove the lemma: To clarify, we believe the reviewer’s question can be rephrased as follows: Since the hockey stick divergence for $M$ is $\max_t [ Pr_{x ~ M(D)} [x \geq t] - e^\epsilon Pr_{x ~ M(D’)}[x \geq t]]$, why does it suffice to bound $Pr_{x ~ M(D)} [x \geq t]$ by $Pr_{x ~ M’(D)} [x \geq t]$ and ignore the term depending on $D’$? The answer is that both $M(D’)$ and $M’(D’)$ are $N(0, \sigma^2)$, so the term depending on $D’$ is the same for both mechanisms. The proof can definitely be made more clear even after fixing the typos and we will rewrite it accordingly in the revision.

---

> > ### Comment · Reviewer_aP5d · 2023-11-21
> >
> > Thanks for the clarification. I appreciate the authors' efforts in making such a seemingly computationally infeasible analysis practical and the frankness in discussing the limitations in detail. I have increased the score accordingly.

---

> > > ### Author Response · Authors · 2023-11-22
> > > **Thanks to the reviewer**
> > >
> > > Thanks to the reviewer for raising the score! We are happy to hear of the reviewer's appreciation of the discussion.

---

### Official Review · Reviewer_5cQP · 2023-11-01

**Soundness:** 3 good
**Presentation:** 2 fair
**Contribution:** 3 good
**Rating:** 6
**Confidence:** 2

**Summary:**

This paper presented an algorithm for calculating tight privacy guarantees of a generic matrix mechanism. The paper opens with a conditional composition theorem which improves the analysis when the noises are correlated. Building on the first theorem, the paper then derives privacy guarantees for the matrix mechanism under uniform sampling. The paper also includes a case study where the algorithm is applied to the binary tree mechanism, backed by experimental evidence.

**Strengths:**

1. The theoretical results look novel and sound. The proposed algorithm is also demonstrated effectively in experiments.
2. The conditional composition theorem could be potentially useful for analyzing general DP mechanisms that involves correlated noise.

**Weaknesses:**

This paper could be more easy to understand if the author could define the matrix mechanism formally in the beginning with some intuitive interpretation of each variable.

**Questions:**

None

---

> ### Author Response · Authors · 2023-11-18
> **Response to reviewer 5cQP**
>
> Thanks to the reviewer for their supportive review.
>
> **"This paper could be more easy to understand if the author could define the matrix mechanism formally in the beginning with some intuitive interpretation of each variable.”**
>
> Thanks for the suggestion on improving the clarity of the paper. To help the reader understand the matrix mechanism, we will discuss two common instances of the matrix mechanism in Section 1.2, where it is first formally defined. Specifically, we will:
> + Move the discussion in Section 5 about the connection between the binary tree mechanism and the matrix mechanism to Section 1.2.
> + Make explicit the connection between the workload matrix A and optimization algorithms such as DP-SGD/DP-FTRL.
> In particular, we will say when A is the all-ones lower-triangular matrix, and x is a stream of gradients computed in DP-FTRL then the rows of Ax are the sums of gradients seen so far. Then, the ith iterate in DP-FTRL is the initialization point plus ith row of Ax+Bz, i.e. the initialization plus the noisy sum of the first i gradients seen, and if C = I, we recover independently noising each gradient / DP-SGD.
>
> (This is a partial duplicate of a response to reviewer 9Vpe)

---

### Official Review · Reviewer_FAAy · 2023-11-05

**Soundness:** 3 good
**Presentation:** 3 good
**Contribution:** 3 good
**Rating:** 8
**Confidence:** 3

**Summary:**

This paper proposes an algorithm to improve the privacy analysis for general matrix mechanisms, a major building block behind DP-FTRL. The paper first proposes a “conditional composition theorem”; then the paper characterizes "high-probability PLD" of matrix mechanisms through Mixture of Gaussian mechanisms.

**Strengths:**

This work is important for the further development of DP-FTRL. The idea of characterizing the worst-case of matrix mechanism through MoG is interesting. The experiment shows improved privacy-utility tradeoff for DP-FTRL.

**Weaknesses:**

I don't see major problem with this paper.

However, I did not check the details of the proof (though the high-level idea in the maintext makes sense to me).

**Questions:**

Can the author comment about the novelty of Theorem 3.1? It seems to be a simple extension of Lemma A.1 in [1]?

[1] Cohen, Edith, and Xin Lyu. "The Target-Charging Technique for Privacy Accounting across Interactive Computations." NeurIPS (2023)

---

> ### Author Response · Authors · 2023-11-18
> **Response to reviewer FAAy**
>
> Thank you for the supportive review. Below we address the question of the reviewer.
>
> **Can the author comment about the novelty of Theorem 3.1? It seems to be a simple extension of Lemma A.1 in [1]?**
>
> Some of the ideas in Theorem 3.1 by itself may not have high-degree of novelty; as the reviewer mentions Cohen and Lyu state something similar, and similar ideas appeared in Erlingsson et al. and Balle et al. (which we mention in the paper). There are some subtleties in correctly extending their Lemma A.1 to compositions of mechanisms instead of a single round mechanism, in a way that fits our use case. In our opinion, the main contributions of Section 3 are
> + Explicitly writing down the general form
> + The framework given by combining Theorem 3.1 with Observation 3.2 to analyze correlated noise mechanisms via a series of independent mechanisms, which we demonstrate is versatile in the later parts of the paper.

---

### Official Review · Reviewer_9Vpe · 2023-11-05

**Soundness:** 3 good
**Presentation:** 3 good
**Contribution:** 3 good
**Rating:** 8
**Confidence:** 4

**Summary:**

This paper considers the problem of privacy amplification due to subsampling or shuffling for the matrix mechanism, which applies to famous algorithms such as binary-tree based algorithm or DP-FTRL. This problem should be interesting to the DP community.

**Strengths:**

This paper proposes MMCC, the first algorithm to analyze privacy amplification for general matrix algorithm. By this method, the paper shows an improved privacy amplification result for the binary-tree based algorithm. Furthermore, the conditional composition theorem, as a side product, should have broader utility. Finally, empirical results reveal a performance improvement due to the better accounting for FTRL.

**Weaknesses:**

Generally speaking, I do not have too much to complain about this paper. The paper considers an interesting question with some nice results. It also lists some reasonable future directions which may help further strengthen the paper. Some minor comments:

1. Although the new privacy amplification applies and improves DP-FTRL, I still believe this paper is more interesting to the DP theory community. Therefore, ICLR might not be the best fit for this paper.
2. As compared to other FFT-based accounting methods, this algorithm should be much more time-consuming. It remains an interesting question to reduce the running time for some specific use cases.
3. I would like to see the authors further improve the clarity of the paper. For now, the paper is not quite friendly to the readers who do not have full background on DP-FTRL or DP accounting. One specific suggestion is to explain how DP-FTRL and binary-tree based algorithm fit into the framework defined in section 1.2.

**Questions:**

Please refer to the section above.

---

> ### Author Response · Authors · 2023-11-18
> **Response to reviewer 9Vpe**
>
> Thank you to the reviewer for their supportive and thoughtful comments. Here we wish to address some of the weaknesses put forth in the review:
>
> **“As compared to other FFT-based accounting methods, this algorithm should be much more time-consuming. It remains an interesting question to reduce the running time for some specific use cases.”**
>
> We agree this is an interesting direction, and one we are currently pursuing as followup work. If the reviewer has not done so already, they may be interested to read Appendix F where we discuss some tricks we use to speed up the calculation at a small cost in accuracy in our implementation. In addition, in Appendix F we discuss how for some special classes of matrices such as the binary tree mechanism and banded Toeplitz matrices, the computation can be dramatically sped up.
>
> **“I would like to see the authors further improve the clarity of the paper. For now, the paper is not quite friendly to the readers who do not have full background on DP-FTRL or DP accounting. One specific suggestion is to explain how DP-FTRL and binary-tree based algorithm fit into the framework defined in section 1.2.”**
>
> Thank you for the suggestion! To improve Section 1.2, we will:
> + Move the discussion in Section 5 about the connection between the binary tree mechanism and the matrix mechanism to Section 1.2.
> + Make explicit the connection between the workload matrix A and optimization algorithms such as DP-SGD/DP-FTRL.
> In particular, we will say when A is the all-ones lower-triangular matrix, and x is a stream of gradients computed in DP-FTRL then the rows of Ax are the sums of gradients seen so far. Then, the ith iterate in DP-FTRL is the initialization point plus ith row of Ax+Bz, i.e. the initialization plus the noisy sum of the first i gradients seen, and if C = I, we recover independently noising each gradient / DP-SGD.
>
> (This is a partial duplicate of a response to reviewer 5cQp)

---

### Author Response · Authors · 2023-11-18
**Small change in the manuscript**

Thank you to the reviewers for their careful reading of the paper and for the insightful questions and suggestions which will help us greatly improve the quality and readability of the paper. In addition to the changes we have mentioned in our responses to specific concerns raised by reviewers, we want to make the reviewers aware of a change we have made to the manuscript.

We found a bug in the code used to generate Figure 4, which invalidates the result in that figure (the rest of the figures/results in the paper are unaffected). In the current Figure 4, we had handcrafted a factorization $A=BC$ to  show that “every-round” sampling and MMCC enable better amplification than the [banded DP-MF paper](https://arxiv.org/abs/2306.08153)’s sampling scheme + amplification analysis due to more randomness in the sampling. However, in the light of the bug, the factorization we handcrafted is suboptimal. But the good news is that the natural binary tree mechanism from Kairouz et al. that we consider in the rest of the paper, enables us to demonstrate the same phenomenon that we initially intended. We have updated Figure 4 and Section 6.3 (and text in the paper referring to this figure/section) appropriately.

We have already uploaded an updated pdf reflecting this change, so the reviewers can make an informed decision. We still plan to make a further revision to incorporate the changes mentioned in our reviewer-specific comments.

---

### Meta-Review · Area_Chair_tM6f · 2023-12-06

**Metareview:**

This paper introduces the first algorithm which analyzes privacy amplification for general matrix mechanisms. As a main technical contribution, a conditional composition theorem is derived that enables to derive new bounds for privacy amplification by subsampling with with correlated noise. The paper includes a case study on the binary-tree mechanism with supportive empirical evidence demonstrating performance improvements.

The methods derived in this paper, especially the  conditional composition theorem, may be beneficial for analyzing privacy amplification in subsampling/shuffling also for other algorithms.
The paper contains a detailed comparison with related works, and a discussion of the limitations of the approach.

Please us the front from the ICLR template in the final version.

**Justification For Why Not Higher Score:**

The paper make a solid theoretical contribution that allows gaining further understanding of matrix mechanisms in DP training and could spur future work.  However, in light of certain limitations that might restrict the practical applications for now, I think it's most suitable to feature this paper as a spotlight instead of an oral presentation.

**Justification For Why Not Lower Score:**

The derived technique could have application in other settings with correlated noise/updates.

---

### Decision · Program_Chairs · 2024-01-16

Accept (spotlight)